# GraphEditor: An Efficient Graph Representation Learning and Unlearning Approach

## Abstract

As graph representation learning has received much attention due to its widespread applications, removing the effect of a specific node from the pre-trained graph representation learning model due to privacy concerns has become equally important. However, due to the dependency between nodes in the graph, graph representation unlearning is notoriously challenging and still remains less well explored. To fill in this gap, we propose GraphEditor, an efficient graph representation *learning* and *unlearning* approach that supports node/edge deletion, node/edge addition, and node feature update for linear-GNN. Compared to existing unlearning approaches, GraphEditor requires neither retraining from scratch nor of all data presented during unlearning, which is beneficial for the settings that not all the training data are available to retrain. Besides, since GraphEditor is exact unlearning, the removal of all the information associated with the deleted nodes/edges can be guaranteed. Empirical results on real-world datasets illustrate the effectiveness of GraphEditor for both node and edge unlearning tasks. The code can be found in supplementary.

## 1 Introduction

In recent years, graph representation learning has been recognized as a fundamental learning problem and has received much attention due to its widespread use in various domains, including social network analysis Kipf & Welling (2017); Hamilton et al. (2017), traffic prediction Cui et al. (2019); Rahimi et al. (2018), knowledge graphs Wang et al. (2019a;b), and recommender systems Berg et al. (2017); Ying et al. (2018). However, due to the increasing concerns on data privacy, removing the effect of a specific data point from the pre-trained model has become equally important. Recently, "*Right to be forgotten*" Wikipedia contributors (2021) empowers the users the right to request the organizations or companies to have their personal data be deleted in a rigorous manner. For example, when Facebook users deregister their account, users not only can request the company to permanently delete the account's profiles from the social network, but also require the company to eliminate the impact of the deleted data on any machine learning model trained based on the deleted data, which is known as machine unlearning Bourtoule et al. (2021).

One of the most straightforward unlearning approaches is to retrain the model from scratch using the remaining data, which could be computationally prohibitive when the dataset size is large or infeasible if not all the data are available to retrain. Recently, many efforts have been made to achieve efficient unlearning, which can be roughly classified into *exact unlearning* and *approximate unlearning*, each of which has its own limitations. ***Exact unlearning***: Bourtoule et al. (2021) proposes to randomly split the original dataset into multiple disjoint shards and train each shard model independently. Upon receiving a data deletion request, the model provider only needs to retrain the corresponding shard model. Chen et al. (2021) extends Bourtoule et al. (2021) by taking the graph into consideration for data partition. However, splitting too many shards could hurt the model performance due to the data heterogeneity and lack of training data for each shard model Ramezani et al. (2021). On the other hand, too few shards result in retraining on massive data, which is computationally prohibitive; ***Approximate unlearning***: Guo et al. (2020); Chien et al. (2022) proposes to approximate the unlearned model using first-order Taylor-expansion, Golatkar et al. (2020) proposes to fine-tune with Newton's method on the remaining data, and Wu et al. (2020a) proposes to transfer the gradient computed at one weight to another and retrain the model from scratch with lower computational cost. Since approximate unlearning lacks guarantee on whether all information associated with the deleted data are eliminated,

these methods require injecting random noise, which can significantly hurt the model performance. Employing graph representation unlearning is even more challenging due to the dependency between nodes that are connected by edges. We not only need to remove the information related to the deleted nodes, but also need to update its impact on neighboring remaining nodes of multi-hops. Since most of the existing unlearning methods only support data deletion, extending their application to graphs is non-trivial. Motivated by the importance and challenges of graph representation unlearning, we aim at answering the following two questions:

*Q1: Can approximate unlearning methods remove all information related to the deleted data?* To verify this, we introduce "*deleted data replay test*" to validate the effectiveness of unlearning in Section 5. Specifically, we add an extra-label category and change all deleted nodes to this extra-label category. To better distinguish deleted nodes from others, an extra binary feature is appended to all nodes and set the extra binary feature as "1" for the deleted nodes and as "0" for other nodes. We first pre-train the model on the dataset with extra label and feature, then we evaluate the effectiveness of unlearning method by comparing the number of the deleted nodes that are predicted as the extra-label category before and after the unlearning process. Intuitively, an effective unlearning method should unlearn all the knowledge related to the additional category and binary feature, a model after unlearning should never predict a node as the additional category. However, according to our observation, approximate unlearning fails to remove all information related to the deleted data, which motivates us to design an efficient exact graph representation unlearning method.

*Q2: If not, can we design an efficient exact graph representation unlearning method?* We propose an exact graph learning and unlearning algorithm GRAPHEDITOR which can efficiently update the parameters with provable low time complexity. GRAPHEDITOR not only supports node/edge deletion, but also node/edge addition and node feature update. The key idea of GRAPHEDITOR is to reformulate the ordinary GNN training problem as an alternative problem with the closed-form solution. Upon receiving a deletion request, GRAPHEDITOR takes the closed-form solution as input and quickly updates the model parameters only based on a small fraction of nodes in the neighborhood of the deleted node/edge. Comparing to retraining from the scratch, GRAPHEDITOR only requires less data with a single step of computation, which is more suitable for *the online setting that requires the model provider to immediately get the unlearned model* or *not all the training data are available to retrain.* Comparing to existing exact unlearning methods GRAPHERASER Chen et al. (2021), GRAPHEDITOR enjoys a better performance since the unlearned model does not suffer from data heterogeneity and lack of training data on each shard model. Comparing to approximate unlearning method INFLUENCE Guo et al. (2020) and FISHER Golatkar et al. (2020), GRAPHEDITOR guarantees removing all information related to deleted nodes/edges and does not require integrating differential privacy noise to prevent information leakage after unlearning.

**Contributions.** We summarize our contributions as follows: ① We introduce "*deleted data reply test*" to validate the effectiveness of unlearning methods and illustrate the insufficiency of approximate unlearning methods for removing all information related to the deleted nodes/edges. ② We introduce a graph representation learning and unlearning approach GRAPHEDITOR on linear-GNNs, which supports node/edge deletion, node/edge addition, and node feature update. ③ To improve the scalability and expressiveness, we introduce subgraph sampling and the non-linearity extension of GRAPHEDITOR. ④ Empirical studies on real-world datasets that illustrates its effectiveness.

## 2 RELATED WORKS

**Exact machine unlearning.** Exact unlearning aims to produce the performance of the model trained without the deleted data. The most straightforward way is to retrain the model from scratch, which is in general computationally demanding, except for some model-specific or deterministic problems such as SVM Cauwenberghs & Poggio (2001), K-means Ginart et al. (2019), and decision tree Brophy & Lowd (2021). Recently, efforts have been made to reduce the computation cost for general gradient-based training problems. For example, Bourtoule et al. (2021) proposes to split the dataset into multiple shards and train an independent model on each data shard, then aggregate their prediction during inference. The data partition schema allows for an efficient retrain of models on a smaller fragment of data. However, the model performance suffers because each model has fewer data to be trained on and data heterogeneity can also deteriorate the performance. Besides, GRAPHERASER Chen et al. (2021) extends Bourtoule et al. (2021) to graph-structured

data by proposing a graph partition method that can preserve the structural information as much as possible and weighted prediction aggregation for inference. Ullah et al. (2021) proposes to train the model using mini-batch SGD and save the model parameters at each iteration. When receiving the deletion requests, retraining only starts at the iteration that deleted data first time appears. Neel et al. (2020); Ullah et al. (2021); Sekhari et al. (2021) study the unlearning from the generalization theory perspective, which is not the main focus of this paper.

**Approximate machine unlearning.** INFLUENCE Guo et al. (2020) proposes to unlearn by removing the influence of the deleted data on the model parameters. Formally, let $\mathcal{D}_d \subset \mathcal{D}$ denote the deleted subset of training data, $\mathcal{D}_r = \mathcal{D} \setminus \mathcal{D}_d$ denote the remaining data, $\mathcal{L}(\mathbf{w})$ is the objective function, and $\mathbf{w}$ is the model parameters before unlearning. Then, INFLUENCE unlearn by $\mathbf{w}^u = \mathbf{w} + \mathbf{H}_r^{-1}\mathbf{g}_d$, which is derived from the first-order Taylor approximation on gradient, where $\mathbf{w}^u$ is the parameters after unlearning, $\mathbf{H}_r = \nabla^2\mathcal{L}(\mathbf{w}, \mathcal{D}_r)$ is the Hessian computed on the remaining data, and $\mathbf{g}_d = \nabla\mathcal{L}(\mathbf{w}, \mathcal{D}_d)$ is the gradient computed on the deleted data. To mitigate the potential information leakage, INFLUENCE utilizes a perturbed objective function $\mathbf{L}(\mathbf{w}) + \mathbf{b}^\top \mathbf{w}$, where $\mathbf{b}$ is the random noise. Guo et al. (2020) requires the objective function be *i.i.d.*, extending its application on graph is non-trivial because nodes in graph are *non-i.i.d* due to node dependency (details in Appendix E). Chien et al. (2022) extends the analysis of Guo et al. (2020) to graph. A similar idea is explored in Wu et al. (2022). FISHER Golatkar et al. (2020) performs Fisher forgetting by taking a single step of Newton's method on the remaining training data, then performing noise injection to model parameters to mitigate the potential information leaking. The model parameters after unlearning is given by $\mathbf{w}^u = \mathbf{w} - \mathbf{H}_r^{-1}\mathbf{g}_r + \mathbf{H}_r^{-1/4}\mathbf{b}$, where $\mathbf{H}_r = \nabla^2\mathcal{L}(\mathbf{w}, \mathcal{D}_r)$ is Hessian and $\mathbf{g}_r = \nabla\mathcal{L}(\mathbf{w}, \mathcal{D}_r)$ is gradient computed on the remaining data $\mathcal{D}_r$, and $\mathbf{b}$ is the random noise. Golatkar et al. (2021) generalizes the idea to deep neural networks by assuming a subset of training samples are never forgotten, which can be used to pre-train a neural network as feature extractor, and only unlearn the last layer. Wu et al. (2020a) proposes to save all the intermediate weight parameters $\mathbf{w}_t$ and gradients $\nabla\mathcal{L}(\mathbf{w}_t, \mathcal{D})$ during training. Then, these information will be used to efficiently estimate the optimization path of strongly convex and smooth objective function after unlearning, which results in very limited applications. Khan & Swaroop (2021) proposes knowledge-adaptation priors to reduce the cost of retraining by enabling adaptation for a wide variety of tasks and models. Similar ideas are explored in Ginart et al. (2019) for K-means and Wu et al. (2020b) for logistic regression. Wang et al. (2021a) observes that different channels have a varying contribution to different categories in image classification. Inspired by this observation, Wang et al. (2021a) proposes to quantize the class discrimination of channels and prune the most relevant channel of the target category to unlearn its contribution to the model. Izzo et al. (2021) propose approximate data deletion method, which has a time complexity that is linear in the dimension of the deleted data and is independent of the size of the dataset. Fu et al. (2022); Nguyen et al. (2022) study Bayesian inference unlearning, which is different from the neural network unlearning that we focused on.

## 3 PRELIMINARIES ON GRAPH REPRESENTATION UNLEARNING

**Problem setup.** Given a graph $\mathcal{G}(\mathcal{V}, \mathcal{E})$ with $N = |\mathcal{V}|$ nodes as input, let us suppose each node $v_i \in \mathcal{V}$ is associated with node feature vector $\mathbf{h}_i^{(0)}$. Let $\mathbf{A}, \mathbf{D} \in \mathbb{R}^{N \times N}$ denote the adjacency matrix and its degree matrix, and the normalized propagation matrix is defined as $\mathbf{P} = \mathbf{D}^{-1/2}\mathbf{A}\mathbf{D}^{-1/2}$. For ease of exposition, we take semi-supervised node classification as a running example, where a subset of nodes $\mathcal{V}_{\text{train}} \subset \mathcal{V}$ are labeled, our goal is to predict the label for the rest nodes $\mathcal{V} \setminus \mathcal{V}_{\text{train}}$ using the information of the labeled nodes. Please notice that GRAPHEDITOR can also be applied to link prediction task for edge unlearning, which will be discussed in details in the appendix.

**Graph neural network.** The feed-forward rule in graph neural network (GNN) is defined as $\mathbf{H}^{(\ell)} = \sigma(\mathbf{P}\mathbf{H}^{(\ell-1)}\mathbf{W}^{(\ell)})$, where $\sigma(\cdot)$ is non-linear activation function, $\mathbf{H}^{(\ell)}$ denotes the hidden representation at the $\ell$-th layer. Then, a linear classifier is applied to the final layer node representation $\mathbf{H}^{(L)}$ for prediction. Although GNNs have become the de-facto tool for graph representation learning, employing unlearning strategies on the ordinary GNNs is non-trivial. This is because how to rigorously verify data removal guarantee in non-linear models is an open problem and non-trivial to verify empirically. Recently, linear-GNNs are proposed to remove non-linearities and only use a single-weight matrix in the neural architecture. For example, SGC Wu et al. (2019) proposes to compute the node representation by $\mathbf{H}^{(L)} = \mathbf{P}^L\mathbf{H}^{(0)}$. By linearizing the GNNs, these methods not only enjoy a faster training speed but also allow us rigorously theoretically and empirically

verify whether the information has been perfectly unlearned. Although lack of non-linearity, recent studies Wei et al. (2022); Wang & Zhang (2022) shows that linear-GNNs are almost as expressive as its non-linear counterparts (details in Appendix F). Motivated by the advantages of linear-GNNs, we will first illustrating our idea on linear-GNNs in Section 4.1 and then introduce its non-linearity extension in Section 4.4. We will rigorously test whether the information is perfectly unlearned on linear-GNNs and also demonstrate the possibility to use GRAPHEDITOR with non-linear GNNs.

**Challenges in graph unlearning.** Graph representation unlearning is challenging for the following three main reasons: ① *High computation cost.* Existing unlearning methods suffer from high computation cost. To see this, let us suppose we are training logistic regression via gradient descent, i.e., $f(\mathbf{w}) = -\sum_{i=1}^{N} y_i \log \mu_i + (1 - y_i) \log(1 - \mu_i)$, where $\mu_i = \sigma(\mathbf{w}^\top \mathbf{x}_i)$ is the prediction and $y_i \in \{0, 1\}$ is the ground truth label. If unlearn by re-training from scratch, it takes $\mathcal{O}(dNE)$ time complexity to unlearn a single data point, where $N$ is the number of data points, $d$ is feature dimension, and $E$ is the number of epochs during training, which is infeasible if the deletion request needs to be completed immediately. Although approximate unlearning methods can alleviate the computation burden to some extent, the computation cost is still linear with respect to the number of nodes $N$. For example, INFLUENCE and FISHER require $\mathcal{O}(Nd)$ to compute gradient $\nabla f(\mathbf{w}) = \mathbf{X}^\top (\boldsymbol{\mu} - \mathbf{y})$ and $\mathcal{O}(Nd^2)$ to compute Hessian $\nabla^2 f(\mathbf{w}) = \mathbf{X}^\top \text{diag}(\boldsymbol{\mu} \cdot (1 - \boldsymbol{\mu})) \mathbf{X}$, which could scale poorly on the large-scale datasets. ② *Non-triviality of extension to graph domain.* Most existing unlearning methods only support data deletion, however, graph representation unlearning also requires updating the effect of the deleted nodes to its neighborhood due to the convolution operation on graph.

For example, as shown in Figure 1, let us suppose our goal is to unlearn the effect of node $v_1$ on a pre-trained 1-layer GCN. After removing node $v_1$, the node representation of node $\{v_2, v_3, v_4\}$ are also affected due to the change of edge weight $\alpha_{ij}$ and the deletion of node $v_1$'s feature. Therefore, a proper graph representation unlearning algorithm not only need to remove the effect of node $v_1$ (which can be achieved by using INFLUENCE and FISHER), but also require to be capable of updating the effect of node $\{v_2, v_3, v_4\}$ on model parameters (which is not supported by most unlearning methods). ③ *Lack of data removal guarantee.* Although approximate unlearning methods are more efficient than retraining from scratch, the removal of all information related to the deleted data is not guaranteed, in which we validate this by "*deleted data replay test*" in Section 5. Intuitively, the above observation makes sense because the output of approximate unlearning is not necessarily equivalent to the result of exact unlearning. Furthermore, most approximate unlearning algorithms seek to prove the approximately unlearned model is close to an exactly retrained model Wu et al. (2020a); Aldaghri et al. (2021); Izzo et al. (2021). However, it has been pointed out by Thudi et al. (2021); Guo et al. (2020) that we cannot infer "*whether the data have been deleted*" solely from "*the closeness of the approximately unlearned and exactly retrained model in the parameter space*". In fact, Thudi et al. (2021) shows that one can even unlearn the data without modifying the parameters. Therefore, it is important to show from the algorithm itself that the sensitive information can be perfectly removed, which is lacking in most approximate unlearning methods due to the approximation process. To overcome the above challenges, we propose GRAPHEDITOR that enjoys a low computation cost with data removal guarantees.

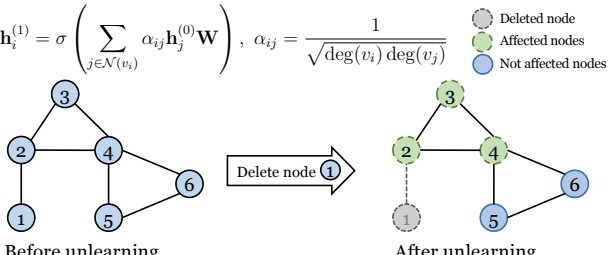

$$\mathbf{h}_i^{(1)} = \sigma\left(\sum_{j \in \mathcal{N}(v_i)} \alpha_{ij} \mathbf{h}_j^{(0)} \mathbf{W}\right), \quad \alpha_{ij} = \frac{1}{\sqrt{\deg(v_i)\deg(v_j)}}$$

Figure 1: An illustration of how output of a 1-Layer GCN is affected after deleting the node $v_1$.

## 4 GRAPHEDITOR

In this section, we first introduce the graph representation learning and unlearning under the notation of linear-GNN in Section 4.1 and Section 4.2, respectively. Then, we introduce the subgraph sampling-strategy to lower the computation cost in Section 4.3 and introduce application of using GRAPHEDITOR with multi-layer GNNs in Section 4.4. We consider both the node unlearning (discussed in main text) and edge unlearning (deferred to appendix). We summarize the full unlearning process of GRAPHEDITOR in Algorithm 1, which consists of three main functions: `find_W()`, `remove_data()`, and `add_data()`.

**Algorithm 1** GRAPHEDITOR (Numpy-like pseudo-code)

```
# Input: X as the output of GNNs, Y as the label matrix
# (Before unlearning) Compute the closed-form solution
>>> S, W = find_W(X, Y)
def find_W(X, Y, reg=0):
    XtX = X.T@X + reg*numpy.eye(X.shape[0]), S = numpy.linalg.inv(XtX), W = S@X.T@Y
    return S, W
# (GraphEditor) Step 1: Delete information
>>> S, W = remove_data(X[V_rm ∪ V_upd], Y[V_rm ∪ V_upd], S, W)
def remove_data(X, Y, S, W):
    I = numpy.eye(X.shape[0])
    A = S@X.T, B = numpy.linalg.inv(I - X@S@X.T), C = Y - X@W, D = X@S
    return S + A@B@D, W - A@B@C
# (GraphEditor) Step 2: Update information. X̃ computed on updated graph.
>>> S, W = add_data(X̃[V_upd], Y[V_upd], S, W)
def add_data(X, Y, S, W):
    I = numpy.eye(X.shape[0])
    A = S@X.T, B = numpy.linalg.inv(I + X@S@X.T), C = Y - X@W, D = X@S
    return S - A@B@D, W + A@B@C
# (Optional) Fine-tune W using cross-entropy loss
```

## 4.1 GRAPH REPRESENTATION LEARNING ON LINEAR-GNN

Instead of training ordinary GNNs by optimizing the cross-entropy loss, we propose to first formulate the ordinary GNN training as a linear GNN training with Ridge regression as the objective, which can be efficiently solved by the closed-form solution. More specifically, we first solve the following Ridge regression problem $\mathcal{L}_{\text{Ridge}}(\mathbf{W}; \mathbf{X}, \mathbf{Y}) = \|\mathbf{X}\mathbf{W} - \mathbf{Y}\|_{\text{F}}^2 + \lambda\|\mathbf{W}\|_{\text{F}}^2$, where $\mathbf{Y} \in \mathbb{R}^{N \times d_y}$ is the zero-one label matrix and $\mathbf{X} \in \mathbb{R}^{N \times d_x}$ is the node representation matrix for linear-GNN (e.g., for SGC we have $\mathbf{X} = \mathbf{P}^L \mathbf{H}^{(0)}$). The closed-form solution for the above objective function is $\mathbf{W}_\star = \arg\min_{\mathbf{W}} \mathcal{L}_{\text{Ridge}}(\mathbf{W}; \mathbf{X}, \mathbf{Y}) = \mathbf{S}_\star \mathbf{X}^\top \mathbf{Y}$, where $\mathbf{S}_\star = (\mathbf{X}^\top \mathbf{X} + \lambda \mathbf{I})^{-1}$ is the inversed correlation matrix. After training, we cache both $\mathbf{S}_\star \in \mathbb{R}^{d_x \times d_x}$ and $\mathbf{W}_\star \in \mathbb{R}^{d_x \times d_y}$ for efficient unlearning. Please refer to `find_W()` for details. To boost model performance, we can take the closed-form solution as initialization and fine-tune using cross-entropy loss with a small number of iterations. The time complexity of computing an exact unlearning solution with closed-form solution is $\mathcal{O}(N d_x^2 + N d_x d_y + d_x^2 d_y)$, which makes retraining on large-scale dataset computationally prohibitive due to linear dependency with respect to the graph size $N$. In the next section, we show that GRAPHEDITOR achieves efficient graph unlearning with computation cost independent of graph size, which makes it suitable for unlearning on large graphs.

## 4.2 GRAPH REPRESENTATION UNLEARNING ON LINEAR-GNN

Let us suppose node $v_i$ is to be deleted. In node unlearning, we not only need to unlearn node $v_i$'s features but also need to unlearn its connection with other nodes. Formally, let $\mathcal{G}_u^{\text{node}}(\mathcal{V}_u^{\text{node}}, \mathcal{E}_u^{\text{node}})$ denote the graph with node $v_i$ and all its associated edges are removed, where $\mathcal{V}_u^{\text{node}} = \mathcal{V} \setminus \{v_i\}$ and $\mathcal{E}_u^{\text{node}} = \mathcal{E} \setminus \{(v_i, v_j) \mid v_j \in \mathcal{N}(v_i)\}$. The model after node unlearning is expected to produce the same performance as the model trained on $\mathcal{G}_u^{\text{node}}$. The key idea of GRAPHEDITOR is to leverage the obtained closed-form solution to first efficiently "*remove*" the effect of the deleted nodes on weight parameters by `remove_data()`, then "*update*" the effect of the neighboring nodes of the deleted nodes on weight parameters by `add_data()`. Before delving into the details of algorithm, let us first take a closer look at the key factors that affect the weight parameters after node deletion.

**Lemma 1** (Key factors that affect weight parameters). *The optimal weight parameters $\mathbf{W}_\star^u$ after removing node $v_i$ is affected by two factors:* ① *"node representation removal" caused by removing node set $\mathcal{V}_{rm} = \{v_i\}$;* ② *"node representation update" due to the inner dependency between nodes in the graph, where the affected node set are all nodes that has shortest path distance (SPD) smaller than $2L$ to nodes in $\mathcal{V}_{rm}$, i.e., $\mathcal{V}_{upd} = \{v_j \mid SPD(v_i, v_j) \leq 2L, \forall v_j \in \mathcal{V}, \forall v_i \in \mathcal{V}_{rm}\}$.*

The above lemma shows that node-set $\mathcal{V}_{\text{rm}}$ and $\mathcal{V}_{\text{upd}}$ are the key factors that affect the weight parameters. Therefore, we propose GRAPHEDITOR to first remove the effect of all node in $\mathcal{V}_{\text{rm}} \cup \mathcal{V}_{\text{upd}}$ on the optimal weight parameters $\mathbf{W}_\star$, then update the effect of $\mathcal{V}_{\text{upd}}$ to derive the optimal weight $\mathbf{W}_\star^u$. More specifically, our first step is to remove the effect of $\mathcal{V}_{\text{rm}} \cup \mathcal{V}_{\text{upd}}$ by `remove_data()`. Let $\mathbf{X}_{\text{rm}} = \mathbf{X}[\mathcal{V}_{\text{rm}} \cup \mathcal{V}_{\text{upd}}], \mathbf{Y}_{\text{rm}} = \mathbf{Y}[\mathcal{V}_{\text{rm}} \cup \mathcal{V}_{\text{upd}}]$ denote the subset of matrix $\mathbf{X}, \mathbf{Y}$ with row

indexed by $\mathcal{V}_{\text{rm}} \cup \mathcal{V}_{\text{upd}}$. Then, given the initial solution $\mathbf{S}_\star$ and $\mathbf{W}_\star$, we first update the inversed correlation matrix as $\mathbf{S}_{\text{rm}} = \mathbf{S}_\star + \mathbf{S}_\star \mathbf{X}_{\text{rm}}^\top [\mathbf{I} - \mathbf{X}_{\text{rm}} \mathbf{S}_\star \mathbf{X}_{\text{rm}}^\top]^{-1} \mathbf{X}_{\text{rm}} \mathbf{S}_\star$ and update the optimal solution by $\mathbf{W}_{\text{rm}} = \mathbf{W}_\star - \mathbf{S}_\star \mathbf{X}_{\text{rm}}^\top [\mathbf{I} - \mathbf{X}_{\text{rm}} \mathbf{S}_\star \mathbf{X}_{\text{rm}}^\top]^{-1} (\mathbf{Y}_{\text{rm}} - \mathbf{X}_{\text{rm}} \mathbf{W}_\star)$. Then, our next step is to update the effect of $\mathcal{V}_{\text{upd}}$ on the weight parameters by `add_data()`. To achieve this, we first compute the updated node representation $\tilde{\mathbf{X}}$ on the graph without the deleted nodes. Let $\mathbf{X}_{\text{upd}} = \tilde{\mathbf{X}}[\mathcal{V}_{\text{upd}}], \mathbf{Y}_{\text{upd}} = \mathbf{Y}[\mathcal{V}_{\text{upd}}]$ denote the subset of matrix $\tilde{\mathbf{X}}, \mathbf{Y}$ with row indexed by $\mathcal{V}_{\text{upd}}$. Then, we update the inversed correlation matrix by $\mathbf{S}_{\text{upd}} = \mathbf{S}_{\text{rm}} - \mathbf{S}_{\text{rm}} \mathbf{X}_{\text{upd}}^\top [\mathbf{I} + \mathbf{X}_{\text{upd}} \mathbf{S}_{\text{rm}} \mathbf{X}_{\text{upd}}^\top]^{-1} \mathbf{X}_{\text{upd}} \mathbf{S}_{\text{rm}}$ and update the optimal solution by $\mathbf{W}_{\text{upd}} = \mathbf{W}_{\text{rm}} + \mathbf{S}_{\text{rm}} \mathbf{X}_{\text{upd}}^\top [\mathbf{I} + \mathbf{X}_{\text{upd}} \mathbf{S}_{\text{rm}} \mathbf{X}_{\text{upd}}^\top]^{-1} (\mathbf{Y}_{\text{upd}} - \mathbf{X}_{\text{upd}} \mathbf{W}_{\text{rm}})$. Notice that GRAPHEDITOR's output is equivalent to the optimal solution $\mathbf{W}_\star^u = \arg\min_{\mathbf{W}} \mathcal{L}_{\text{Ridge}}(\mathbf{W}; \tilde{\mathbf{X}}, \mathbf{Y})$. Besides, since we are using the closed-form solution, we do not have to worry about the information of the deleted nodes in $\mathcal{V}_{\text{rm}}$ might be potentially remained in the weight parameters. The time complexity for graph unlearning is $\mathcal{O}(M^3 + Md_x^2 + Md_x d_y)$, where $M = |\mathcal{V}_{\text{rm}} \cup \mathcal{V}_{\text{upd}}|$. According to time complexity, we know GRAPHEDITOR enjoys a lower computation cost than retraining from scratch if $M < d_x$. When $M$ is large, we could split the removed nodes into multiple small shards and unlearn them one after another, which alleviates the cubic computation dependency on $M$. Since we are using the closed-form solution, the results after unlearning nodes are identical regardless of how many shards we split to unlearn. Besides, since efficient matrix inverse algorithms (e.g., BLAS and LAPACK) have been implemented in Numpy, the practical computation time of GRAPHEDITOR is significantly smaller than re-training or other unlearning methods according to our observations. Details on the correctiveness of GRAPHEDITOR and the time complexity please refer to Appendix C.

**Remark 1** (Node/Edge addition). *The process of node/edge addition is identical to node/edge deletion. For example, we can add nodes by first unlearn the set of all "affected nodes" using the* `remove_data()`, *then perform information update on the "affected nodes" and the "added nodes" use the* `add_data()`. *Here, the "affected nodes" are nodes that have SPD smaller than $2L$ to the "added nodes", which is similar to the node deletion operation. The same applies to edge addition.*

### 4.3 SUBGRAPH SAMPLING FOR BETTER SCALABILITY

Recall from Lemma 1 that the number of nodes in $\mathcal{V}_{\text{upd}}$ grows twice exponentially with respect to the linear-GNN depth, i.e., by letting $D$ as the maximum node degree we have $|\mathcal{V}_{\text{upd}}| \leq |\mathcal{V}_{\text{rm}}| \times D^{2L}$. When the linear GNN is deep, GRAPHEDITOR becomes computational prohibitive even when the deleted node set $\mathcal{V}_{\text{rm}}$ is small, since GRAPHEDITOR's overall computation cost is cubic with respect to $|\mathcal{V}_{\text{upd}} \cup \mathcal{V}_{\text{rm}}|$. To overcome the aforementioned issue, we propose to decouple the receptive field of each node with the GNN depth by extracting the rooted-subgraph for each node in the graph using *shaDow*-subgraph sampler Zeng et al. (2021), and apply the linear GNN to compute the feature representation of the root node on the extracted subgraph. In practice, we can either use the $K$-hop sampling to uniformly sampling the $K$-hop neighbors of root node or we can sample a fixed number of nodes according to the PPR score with respect to the root node.

**Lemma 2** (Affected nodes after sampling). *Let us denote $\mathcal{V}_j^{sg}$ as the set of nodes returned by shaDow-subgraph sampler for root node $v_j$. If using shaDow-subgraph sampler, the affected node set of "node representation update" in Lemma 1 are reduced to $\mathcal{V}_{\text{upd}}^{sg} = \{v_j \mid v_i \in \mathcal{V}_j^{sg}, \ \forall v_j \in \mathcal{V}, \ \forall v_i \in \mathcal{V}_{\text{rm}}\}$*

As shown in Lemma 2, we can reduce the size of update node set to $|\mathcal{V}_{\text{upd}}^{\text{sg}}|$, which is independent of GNN depth. In practice, we find $K$-hop sampling works well on all datasets. When using $K$-hop sampling, the update node set size is reduced to $|\mathcal{V}_{\text{upd}}^{\text{sg}}| \leq |\mathcal{V}_{\text{rm}}| \times D^K$ and we only need to update the representation of a node if its $K$-hop rooted subgraph contains the deleted nodes.

### 4.4 USING GRAPHEDITOR WITH NON-LINEAR MULTI-LAYER GNNS

One of the biggest concern readers might have is the linear-GNN requirement.[1] Extending the existing unlearning methods to non-linear models requires first pre-training multi-layer GNNs as a feature extractor on the public datasets (assume nodes in the public set will never need to be unlearned in the future), then we use the pre-trained multi-layer GNNs to extract the node representation for

---

[1]Please notice that the linearity is not only required by GRAPHEDITOR but also required by most approximate unlearning methods (e.g., Guo et al. (2020); Golatkar et al. (2020); Wu et al. (2020a); Golatkar et al. (2021)) to theoretically show the data removal guarantee. Unless re-training from scratch, how to rigorously show data removal guarantee in non-linear models is still an open problem and is non-trivial to verify empirically.

a linear classifier[2]. During unlearning, we only unlearn the linear classifier without updating the feature extractor because it does not carry any information on the deleted data. Formally, let us define $\mathcal{V}_{\text{public}}, \mathcal{V}_{\text{private}}$ as the nodes in the public and private dataset, $\mathcal{V}_{\text{rm}} \cap \mathcal{V}_{\text{public}} = \emptyset$ and $\mathcal{V}_{\text{rm}} \subset \mathcal{V}_{\text{private}}$, define $f_w \circ g_\theta$ as a multi-layer GNN model with $f_w$ as the final linear classifier and $g_\theta$ as all previous layers. Let us suppose we could pre-train $g_\theta$ on the public dataset nodes $\mathcal{V}_{\text{public}}$, then we use the $g_\theta$ as a feature extractor to extract the node representation $\mathbf{X}$ on private graph node $\mathcal{V}_{\text{private}}$. By doing so, we can apply `find_W()` on $\mathbf{X}$ to learn the linear classifier $f_w$ by GRAPHEDITOR. To unlearn node $\mathcal{V}_{\text{rm}}$, we can first apply `remove_data()` with $\mathbf{X}[\mathcal{V}_{\text{rm}} \cup \mathcal{V}_{\text{upd}}^{\text{sg}}]$ to unlearn the information of $\mathcal{V}_{\text{rm}} \cup \mathcal{V}_{\text{upd}}^{\text{sg}}$ on $f_w$, then compute the node representation $\tilde{\mathbf{X}}$ on the update graph using $g_\theta$, and finally apply `add_data()` with $\tilde{\mathbf{X}}[\mathcal{V}_{\text{upd}}^{\text{sg}}]$ to update the information of $\mathcal{V}_{\text{upd}}^{\text{sg}}$ on $f_w$. Please notice that using GRAPHEDITOR with non-linear GNNs is similar to using GRAPHEDITOR with linear-GNNs, except the original $\mathbf{X}, \tilde{\mathbf{X}}$ are computed using linear-GNNs but here we use non-linear GNNs instead.

## 5 EXPERIMENTS

**Datasets and baselines.** We select *OGB-Arxiv, OGB-Products, Flickr, and Reddit* datasets for node unlearning evaluation, and *OGB-Collab* dataset for edge unlearning evaluation. We compare with approximate unlearning methods INFLUENCE and FISHER on linear-GNNs (by extending their application from the non-structured data to the structured data) and exact unlearning method GRAPHERASER on both linear and non-linear GNNs. Besides, we compare with retraining GCN Kipf & Welling (2017), GraphSAGE Hamilton et al. (2017), GAT Veličković et al. (2017) from scratch.

**Representation computation.** For node unlearning, the node representation is extracted from the sampled rooted subgraph of each node and label reuse trick Wang et al. (2021b) is used for linear-GNNs node classification. For edge unlearning, we first extract the subgraph of the two nodes connected by that edge, then the edge representation is extracted from the intersection of the two subgraphs, common neighbor score Liben-Nowell & Kleinberg (2007) is used for linear-GNNs.

**Overview on experiments.** We measure the success of an unlearning method by two criteria: *whether the information is unlearned* and *whether the unlearning algorithm could deteriorate the model performance*. Please notice that the second criteria is as important as the first criteria. For example, in an extreme case, one can just unlearn by randomly initializing the model but it might significantly hurt the model performance. To validate the above two criteria, we conduct the following experiments: In section 5.1, we conduct deleted data replay test on linear-GNN to test whether an unlearning method could perfectly unlearn the features-labels correlation from the deleted nodes, and whether the unlearning method hurt the model's prediction accuracy. In section 5.2, we compare the similarity between the unlearned model to the re-training from the scratch model on linear-GNN. The two models are expected to be similar to prevent information leakage on the deleted nodes. In section 5.3, we evaluate the efficiency and effectiveness of using GRAPHEDITOR with non-linear GNNs. We compare the accuracy with re-training non-linear GNNs from scratch. Intuitively, a strong unlearning method should unlearn in a short time and produce a similar model performance to re-training. Furthermore, we conduct edge unlearning test in Appendix A.1, ablation study the effect of subgraph sampling size on GRAPHEDITOR for node unlearning in Appendix A.2 and for edge unlearning in Appendix A.3, conduct node addition test in Appendix A.4, compare the prediction confidence on the unlearned node in Appendix A.5, compare the running time in Appendix A.6, and compare linear-GNN with shallow sampler and different multi-layer GNNs in Appendix A.7. Details on datasets, baselines, experiment details are summarized in Appendix B.

### 5.1 DELETED DATA REPLAY TEST

In this experiment, we randomly select 100 nodes from the training set as the deleted nodes and modify their label categories to an extra-label category. An extra binary feature is injected to the node features to help linear-GNNs memorize the correlation between deleted nodes to the extra-label category. To simulate real-world deletion unlearning requests that come one after another, we first uniformly split all deleted nodes into $S \in \{10, 50, 100\}$ shards, then we randomly select one shard without replacement to unlearn at each unlearning iteration and repeat this $S$ times. We measure the success of unlearning by checking whether the information on the deleted nodes is unlearned and

---
[2]Similar assumption are made in Golatkar et al. (2021) for convolutional neural network unlearning.

Table 1: Comparison on the accuracy (before parentheses), number of deleted nodes that are predicted as the extra-label category before and after unlearning (inside parentheses), and wall-clock time.

| Method | | OGB-Arxiv | | | OGB-Products | | |
|---|---|---|---|---|---|---|---|
| | | S=10 | S=50 | S=100 | S=10 | S=50 | S=100 |
| GRAPHEDITOR | Before | 71.77% (70) | 71.77% (70) | 71.77% (70) | 77.63% (83) | 77.63% (83) | 77.63% (83) |
| | After | 71.78% (0) | 71.78% (0) | 71.78% (0) | 77.63% (0) | 77.63% (0) | 77.63% (0) |
| | Time | 10.8 s | 10.9 s | 11.9 s | 46.6 s | 76.9 s | 108.3 s |
| GRAPHEDITOR + Fine-tune | Before | 73.87% (88) | 73.87% (88) | 73.87% (88) | 79.26% (71) | 79.26% (71) | 79.26% (71) |
| | After | 73.86% (0) | 73.86% (0) | 73.86% (0) | 79.26% (0) | 79.25% (0) | 79.25% (0) |
| | Time | 14.7 s | 14.8 s | 14.8 s | 54.6 s | 85.0 s | 117.4 s |
| GRAPHERASER | Before | 69.91% (28) | 69.91% (28) | 69.91% (28) | 63.27% (32) | 63.27% (32) | 63.27% (32) |
| | After | 69.90% (0) | 69.90% (0) | 69.89% (0) | 63.27% (0) | 63.28% (0) | 63.25% (0) |
| | Time | 615.9 s | 1,888.1 s | 2,237.8 s | 15,191.4 s | 39,612.5 s | 46,491.4 s |
| INFLUENCE | Before | 72.99% (93) | 72.99% (93) | 72.99% (93) | 78.05% (63) | 78.05% (63) | 78.05% (63) |
| | After | 72.89% (53) | 72.89% (53) | 72.89% (53) | 78.05% (19) | 78.03% (19) | 78.04% (19) |
| | Time | 62.1 s | 284.7 s | 554.8 s | 151.7 s | 614.2 s | 1,185.7 s |
| FISHER | Before | 72.94% (94) | 72.94% (94) | 72.94% (94) | 78.05% (63) | 78.05% (63) | 78.05% (63) |
| | After | 72.73% (56) | 72.70% (55) | 72.69% (54) | 77.87% (57) | 77.86% (57) | 77.76% (54) |
| | Time | 77.1 s | 364.4 s | 703.5 s | 185.3 s | 791.8 s | 1,528.6 s |

whether the unlearning algorithm could deteriorate the model performance: ① To measure the first criteria, we compare the number of the deleted nodes that is computed as the extra-label category. The prediction of deleted nodes is computed on the graph structure before node deletion, therefore namely "deleted data replay test". Intuitively, a successful unlearning should never predict a node as the extra-label category after the unlearning process. As shown in Table 1, exact unlearning methods GRAPHEDITOR and GRAPHERASER can unlearn all the information because none the deleted nodes are predicted as the extra-label category. However, this is not the case for approximate unlearning methods INFLUENCE and FISHER. ② To measure the second criteria, we report the "accuracy" before and after unlearning. A strong unlearning method should not hurt the model performance, therefore they should produce high accuracy both before and after unlearning.[3] From Table 1, we know that GRAPHERASER, INFLUENCE, and FISHER need to sacrifice their performance to achieve efficient unlearning, therefore they have a lower performance both before and after unlearning. GRAPHERASER has lower performance because of data heterogeneity (data distribution on each shard is different from each other) and lack of training data for each shard model (caused by graph partition). INFLUENCE and FISHER have lower performance because a large regularization term is required to stabilize the Hessian inverse computation and also due to random noises. ③ Moreover, we also compare the running time in Table 1. As the number of shards $S$ increases, the times required by GRAPHEDITOR increases less than baselines. This is because increasing $S$ could also decreases the number of nodes to unlearn at each iteration and the time complexity of GRAPHEDITOR is independent of the dataset size (refer to in Section 4.2). Please notice that this is not the case for baselines because their complexity is always proportional to the dataset size (refer to part three of Section 3). Therefore, GRAPHEDITOR is more efficient than other baseline methods.

## 5.2 COMPARISON TO RETRAINED MODEL

To assess the similarity between the unlearned model and the retrained model, a natural way is to measure the distance between the final activations obtained by the unlearned $\mathbf{W}^u$ and the retrained $\mathbf{W}^r$ models on the deleted nodes and testing set nodes. More specifically, for $\mathcal{B} \in \{\mathcal{V}_{\text{rm}}, \mathcal{V}_{\text{test}}\}$ we compare the distance of final activations as $\mathbb{E}_{v_i \in \mathcal{B}}\big[\|\text{softmax}(\mathbf{x}_i\mathbf{W}^u) - \text{softmax}(\mathbf{x}_i\mathbf{W}^r)\|_2\big]$. The deleted nodes are randomly selected 100 samples from the training set and are unlearned through 10 sequential forgetting requests, each request of size 10. Intuitively, a powerful unlearning algorithm should generate similar final activations to the retrained model on nodes from both the deleted nodes and testing set nodes. Besides, we measure the Euclidean distance between the parameters returned by the unlearning algorithms and the parameters obtained via retraining from scratch. For linear GNN, a small Euclidean distance in the parameter space means the model is likely to have the same predictions. We have the following observations according to Figure 2: ① We observe that GRAPHEDITOR's (both with and without fine-tune) final activation difference and parameter difference is consistently low compared to baselines during the 10 unlearning requests. However, this

---

[3] We are not comparing the difference between the accuracy before and after unlearning because it is not necessarily related to whether the unlearning success but mainly related to the number of data to unlearn.

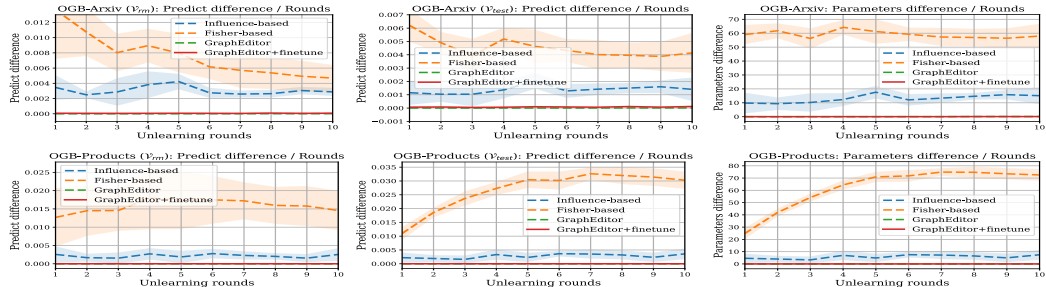

Figure 2: Comparison on the difference of final activation prediction on deleted nodes (1st column) and testing nodes (2nd column) and difference of weight parameters (3rd column).

is not the case for approximate unlearning methods INFLUENCE and FISHER. ② We observe that the final activation differences of approximate unlearning methods on the deleted nodes are consistently larger than the values on the test nodes. Therefore, a malicious third party could potentially identify the deleted nodes from other nodes by comparing its final activation difference.

## 5.3 GRAPHEDITOR WITH NON-LINEAR GNN

In this section, we demonstrate the potential extension of using GRAPHEDITOR with non-linear GNNs. This experiment is conducted under the assumption that a subset of training samples are never forgotten (i.e., public dataset), which can be used to pretrain a neural network as feature extractor, and only unlearn the final linear classifier. To test under this scenario, we randomly split the training set into 90% and 10% for public dataset and private dataset. For GRAPHEDITOR, the feature extractor is pre-trained on the 90% training set nodes, then it is used to extract node representation for all nodes. The extracted representation will be used to train the linear classifier. GRAPHEDITOR is only applied to the linear classifier since the pre-trained feature extractor do not have any information about the deleted nodes. For both re-training and GRAPHERASER, the multi-layer GNNs are trained on all nodes. We select 100 nodes from the 10% private training set with the largest node degree to unlearn. We use 3-layer GAT, GCN, and SAGE with hidden dimension 256, attention head size 8 as the backbone model. We are using the official implementation of GRAPHERASER (detailed setup in Appendix B.3). From Table 2, we observe that: ① GRAPHEDITOR could attain similar accuracy as re-training from scratch but with shorter time, ② GRAPHEDITOR has slightly lower precision than re-training because it has less data to training the feature extractor, ③ the performance of GRAPHERASER is relatively lower than GRAPHEDITOR and re-training due to lack of training data on each shard model, data heterogeneity caused by graph partitioning, and is prone to over-fitting during shard model training phase.

Table 2: Comparison on the accuracy after unlearning and running time on non-linear GNNs.

| | | Retrain | GRAPHEDITOR | GRAPHERASER |
|---|---|---|---|---|
| **OGB-Arxiv** | SAGE | $72.58 \pm 0.17$ (1,211 sec) | $71.13 \pm 0.19$ (19 sec) | $56.32 \pm 0.23$ (224 sec) |
| | GAT | $72.23 \pm 0.18$ (1,425 sec) | $71.01 \pm 0.19$ (21 sec) | $57.16 \pm 0.56$ (301 sec) |
| | GCN | $71.49 \pm 0.27$ (1,089 sec) | $70.89 \pm 0.30$ (18 sec) | $55.90 \pm 0.51$ (277 sec) |
| **OGB-Products** | SAGE | $80.67 \pm 0.37$ (3,156 sec) | $79.00 \pm 0.39$ (56 sec) | $56.53 \pm 0.50$ (11,648 sec) |
| | GAT | $81.42 \pm 0.31$ (3,235 sec) | $79.41 \pm 0.37$ (59 sec) | $62.47 \pm 0.62$ (15,496 sec) |
| | GCN | $79.14 \pm 0.44$ (3,071 sec) | $78.11 \pm 0.48$ (55 sec) | $56.41 \pm 0.49$ (11,298 sec) |
| **Flickr** | SAGE | $53.95 \pm 0.13$ (311 sec) | $52.87 \pm 0.15$ (11 sec) | $43.15 \pm 0.54$ (194 sec) |
| | GAT | $53.64 \pm 0.26$ (328 sec) | $52.42 \pm 0.27$ (13 sec) | $43.14 \pm 0.54$ (243 sec) |
| | GCN | $52.86 \pm 0.13$ (305 sec) | $52.50 \pm 0.16$ (12 sec) | $44.01 \pm 0.53$ (190 sec) |
| **Reddit** | SAGE | $97.03 \pm 0.03$ (3,300 sec) | $96.11 \pm 0.03$ (60 sec) | $88.91 \pm 0.03$ (728 sec) |
| | GAT | $97.10 \pm 0.04$ (3,598 sec) | $96.21 \pm 0.04$ (61 sec) | $87.78 \pm 0.04$ (993 sec) |
| | GCN | $96.24 \pm 0.02$ (3,238 sec) | $95.98 \pm 0.03$ (60 sec) | $87.54 \pm 0.03$ (730 sec) |

## 6 CONCLUSION

We study the problem of graph representation unlearning and propose an exact unlearning algorithm GRAPHEDITOR, which supports node/edge deletion, node/edge addition, and node feature update. GRAPHEDITOR requires neither retraining from scratch nor all data presented during unlearning, and enjoys a guarantee on the removal of all the information associated with the deleted nodes/edges. Extensive experiments on real-world datasets indicate its effectiveness and efficiency.

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

GRAPHEDITOR: AN EFFICIENT GRAPH REPRESENTATION LEARNING AND UNLEARNING APPROACH

**Organization.** In Section A, we provide additional experiment results on both node/edge unlearning and comparison of linear GNNs to ordinary multi-layer non-linear GNNs. More specifically, we conduct edge unlearning test in Appendix A.1, ablation study the effect of subgraph sampling size on GRAPHEDITOR for node unlearning in Appendix A.2 and for edge unlearning in Appendix A.3, conduct node addition test in Appendix A.4, compare the prediction confidence on the unlearned node in Appendix A.5, compare the running time in Appendix A.6, and compare linear-GNN with shallow sampler and different multi-layer GNNs in Appendix A.7. In Section B, we provide details on experiment setup. In Section C, we provide detailed analysis on the computation complexity and correctness of GRAPHEDITOR. In Section D, we provided proof for Lemma 1 and Lemma 2. In Section E we highlight the dependency issue of applying existing unlearning approach to graph structured data. In Section F, we summarize recent theoretical analysis showing that linear-GNN could be almost as expressive as non-linear GNNs. [Code] to reproduce the experiment results can be find from the anonymous repository.

# A  MORE EXPERIMENT RESULTS

## A.1  EDGE UNLEARNING

In this section, we introduce our edge unlearning problem formulation and demonstrate our results. Suppose edge $(v_i, v_j)$ is to be deleted, our goal is to unlearn the connectivity. Let $\mathcal{G}_u^{\text{edge}}(\mathcal{V}, \mathcal{E}_u^{\text{edge}})$ denotes the graph with edge $(v_i, v_j)$ removed, where $\mathcal{E}_u^{\text{edge}} = \mathcal{E} \setminus \{(v_i, v_j)\}$. The model after unlearning is expected to produce the same performance as the model trained on $\mathcal{G}_u^{\text{edge}}$. As shown in Table 3, we compare the model performance (measured by *Hits@50* for edge unlearning), wall-clock time (measured by seconds), and the number of deleted nodes that are predicted as the extra-label category before and after unlearning (reported in the parenthesis), and have the following observations: ① We can observe that different from node-level tasks, the performance of GRAPHEDITOR to baselines are very close. This is potentially due to the nature of OGB-Collab dataset and the feature extraction strategy we used on linear GNN. In fact, we found that the feature extracting strategy plays a more important role in the OGB-Collab dataset, please refer to the next section for more detailed discussions and ablation study results in Table 5. ② Besides, we can observe that the exact unlearning methods GRAPHEDITOR and GRAPHERASER can always unlearn all the information related to the deleted nodes, however, this is not the case for approximate unlearning methods INFLUENCE and FISHER. ③ GRAPHEDITOR is significantly more efficient than other baseline methods, mainly due to its unlearning complexity is independent of the dataset size, which requires less wall-clock time throughout the unlearning process.

Table 3: Comparison on the accuracy (in front of the parentheses), the number of the deleted nodes that are predicted as the extra-label category before and after unlearning (inside the parentheses), and wall-clock time.

| | Method | | S=10 | S=50 | S=100 |
|---|---|---|---|---|---|
| OGB-Collab (Hits@50) | GRAPHEDITOR | Before | 63.45% (97) | 63.45% (97) | 63.45% (97) |
| | | After | 62.69% (0) | 57.42% (0) | 56.34% (0) |
| | | Time | 6.9 s | 13.0 s | 19.4 s |
| | GRAPHEDITOR + Fine-tune | Before | 64.00% (59) | 64.00% (59) | 64.00% (59) |
| | | After | 63.76% (0) | 63.67% (0) | 63.39% (0) |
| | | Time | 8.1 s | 14.2 s | 20.5 s |
| | GRAPHERASER | Before | 63.82% (21) | 63.82% (21) | 63.82% (21) |
| | | After | 63.82% (0) | 63.82% (0) | 63.82% (0) |
| | | Time | 8,990.7 s | 21,586.8 s | 29,871.4 s |
| | INFLUENCE | Before | 63.76% (76) | 63.76% (76) | 63.76% (76) |
| | | After | 63.57% (55) | 63.19% (42) | 63.54% (61) |
| | | Time | 20.8 s | 83.1 s | 159.2 s |
| | FISHER | Before | 64.33% (31) | 64.33% (31) | 64.33% (31) |
| | | After | 63.93% (1) | 63.95% (1) | 63.91% (1) |
| | | Time | 27.4 s | 114.8 s | 230.8 s |

## A.2 Effectiveness of GraphEditor for node unlearning

We study the effectiveness of GraphEditor by comparing it with ordinary GNNs (including GCN and GraphSAGE) and provide an ablation study on the effect of the number of layers per-hop on performance and efficiency. Besides, we also provide experimental results by applying GraphEraser[4] onto ordinary GNNs by splitting the original graph into 8 subgraphs and using mean-aggregation during inference, where the time is reported by the maximum time trained on a single subgraph. We repeat experiment 5 times, each time 100 nodes are randomly selected as deleted nodes from the training set, for node unlearning we randomly split the deleted nodes into 10 sequential forgetting requests of equal size. The results are reported in Table 4, where we denote full-neighbor subgraph as "*Full*", denote subgraph with $K$ neighbors per hop as "*SG (K)*", and denote model with fine-tuning as "*+ FT*". We have the following observation from Table 4: ① adding neighbors per hop not necessarily results in a better model performance on the linear GNN used in GraphEditor, which can be explained by the over-smoothing hypothesis in Cong et al. (2021); Li et al. (2018). ② fine-tuning can bring around 3% of performance-boosting on node classification datasets, which indicates the importance of finetuning. ③ linear GNN can achieve compatible results (even outperform) the ordinary multi-layer non-linear GNN with significantly less computation time, which motivates us to explore better feature engineering tricks for linear GNN as a future direction. ④ GraphEraser suffers performance degradation issue due to the data heterogeneity and lack of training data on each subgraph, which is aligned with our observation on using GraphEraser with linear GNNs as reported in Table 1. Interesting, we found that the performance degradation issue using ordinary GNN is more severe than the linear GNN and the results reported in their original paper Chen et al. (2021). This is potentially due to ordinary non-linear GNNs requires more data for training than linear GNN because of its higher model complexity.

Table 4: Comparison on the effect of subgraph sampling and fine-tuning of GraphEditor on linear-GNNs. Besides, we also compare with re-training multi-layer GNNs from scratch and using GraphEraser for multi-layer GNNs (marked with †).

|  | Method | Accuracy (Before) | Accuracy (After) | Time |
|---|---|---|---|---|
| **OGB-Arxiv** | Full | $69.78 \pm 0.02$ | $69.78 \pm 0.02$ | 3968.4 s |
|  | Full + FT | $74.04 \pm 0.06$ | $74.02 \pm 0.02$ | 3971.6 s |
|  | SG (10) | $71.49 \pm 0.02$ | $71.42 \pm 0.02$ | 6.6 s |
|  | SG (20) | $71.99 \pm 0.02$ | $71.98 \pm 0.02$ | 20.8 s |
|  | SG (50) | $71.51 \pm 0.03$ | $71.52 \pm 0.02$ | 126.7 s |
|  | SG (10) + FT | $73.75 \pm 0.07$ | $73.51 \pm 0.07$ | 10.4 s |
|  | SG (20) + FT | $74.07 \pm 0.07$ | $74.04 \pm 0.07$ | 24.3 s |
|  | SG (50) + FT | $74.29 \pm 0.06$ | $74.26 \pm 0.06$ | 130.0 s |
|  | †GCN | $71.74 \pm 0.29$ | $71.67 \pm 0.26$ | 961.4 s |
|  | †GraphSAGE | $71.49 \pm 0.27$ | $71.41 \pm 0.30$ | 686.6 s |
|  | †GCN + GraphEraser | $66.52 \pm 0.31$ | $66.51 \pm 0.32$ | 137.8 s |
|  | †GraphSAGE + GraphEraser | $65.96 \pm 0.26$ | $65.96 \pm 0.31$ | 107.6 s |
| **OGB-Products** | SG (10) | $76.63 \pm 0.02$ | $76.63 \pm 0.02$ | 47.6 s |
|  | SG (15) | $77.06 \pm 0.03$ | $77.06 \pm 0.02$ | 259.3 s |
|  | SG (20) | $77.11 \pm 0.02$ | $77.12 \pm 0.02$ | 610.8 s |
|  | SG (10) + FT | $79.26 \pm 0.07$ | $79.25 \pm 0.07$ | 54.2 s |
|  | SG (15) + FT | $79.32 \pm 0.06$ | $79.32 \pm 0.06$ | 265.5 s |
|  | SG (20) + FT | $79.51 \pm 0.07$ | $79.52 \pm 0.07$ | 617.1 s |
|  | †GraphSAGE | $78.70 \pm 0.36$ | $78.68 \pm 0.30$ | 12539.5 s |
|  | †GraphSAINT | $79.08 \pm 0.24$ | $79.07 \pm 0.25$ | 7061.1 s |
|  | †ClusterGCN | $78.99 \pm 0.36$ | $79.00 \pm 0.37$ | 11459.8 s |
|  | †GraphSAGE + GraphEraser | $58.99 \pm 0.40$ | $58.87 \pm 0.41$ | 3707.2 s |
|  | †GraphSAINT + GraphEraser | $59.54 \pm 0.41$ | $59.39 \pm 0.39$ | 2271.3 s |
|  | †ClusterGCN + GraphEraser | $59.10 \pm 0.57$ | $59.01 \pm 0.60$ | 3351.7 s |

---

[4]Here, we are using our implementation of GraphEraser by directly applying GCN, GraphSAGE, GraphSAINT, and ClusterGCN onto the graph partitioned by METIS and using mean-average pooling for aggregation. We do this to make sure only the unlearning method is different and other parts are consistent among different unlearning methods (e.g., neural architecture, hyper-parameters). We believe our implementation is general enough and has already captured the leading spirit of GraphEraser, i.e., split data into multiple shards and train a different model on each graph partition. METIS allows us to split the original graph into multiple subgraphs while preserving the original graph structure as much as possible. We would like to point out that the experimental results using official implementations (Table 2) are consistent with our implementation's results and meet our expectations.

### A.3 EFFECTIVENESS FOR EDGE UNLEARNING.

We conduct similar experiment to Appendix A.2 for edge unlearning. We repeat experiment 5 times, each time 100 edges are randomly selected as deleted edges from the training set, for edge unlearning we unlearn through 100 forgetting requests The results are reported in Table 5, where we denote full-neighbor subgraph as "*Full*", denote subgraph with $K$ neighbors per hop as "*SG (K)*", and denote model with fine-tuning as "*+ FT*". We have the following observation from Table 5. ① Addinhg neighbors per hop not necessarily results in a better model performance on the linear GNN used in GRAPHEDITOR, which can be explained by the over-smoothing hypothesis in Cong et al. (2021); Li et al. (2018). ② Fine-tuning can bring very less improvements to link prediction datasets, which is potential because node features are less important in the OGB-Collab dataset, details please refer to here. ③ Linear GNN can achieve compatible results (even outperform) the ordinary multi-layer non-linear GNN with significantly less computation time, which motivates us to explore better feature engineering tricks for linear GNN as a future direction.

Table 5: Comparison on the effect of subgraph sampling and fine-tuning with ordinary GNNs for *edge unlearning*.

| | Method | Hit@50 (Before) | Hit@50 (After) | Time |
|---|---|---|---|---|
| | Full | $63.98 \pm 0.02$ | $63.36 \pm 0.02$ | 91.6 s |
| | Full + FT | $64.68 \pm 0.06$ | $64.68 \pm 0.02$ | 92.1 s |
| | SG (50) | $63.45 \pm 0.02$ | $63.30 \pm 0.02$ | 19.7 s |
| | SG (100) | $63.12 \pm 0.03$ | $63.22 \pm 0.02$ | 28.4 s |
| OGB-Collab | SG (200) | $63.15 \pm 0.02$ | $63.25 \pm 0.02$ | 48.9 s |
| | SG (50) + FT | $63.63 \pm 0.07$ | $63.62 \pm 0.07$ | 20.3 s |
| | SG (100) + FT | $65.52 \pm 0.06$ | $64.45 \pm 0.06$ | 28.9 s |
| | SG (200) + FT | $64.60 \pm 0.07$ | $64.59 \pm 0.07$ | 49.4 s |
| | GCN | $47.14 \pm 1.45$ | $46.99 \pm 1.56$ | $499,367.7$ s |
| | GraphSAGE | $54.63 \pm 1.12$ | $54.49 \pm 1.14$ | $522,571.2$ s |

### A.4 NODE ADDITION TEST

In this experiment, we first randomly select 100 nodes from the training set as the node set $\mathcal{V}_{\text{add}}$ to add, then remove them from the graph, including all edges that are connected to $\mathcal{V}_{\text{add}}$. Similar to the "*deleted node reply test*", extra-label category and binary feature are added to all nodes, where we edit the label of nodes in $\mathcal{V}_{\text{add}}$ to this additional label category, and set the extra feature as "1" for all node in $\mathcal{V}_{\text{add}}$, and set as "0" for all other nodes in $\mathcal{V} \setminus \mathcal{V}_{\text{add}}$. Then, we pre-train our model on the modified dataset. We randomly split the 100 added nodes into $S \in \{10, 50, 100\}$ shards. At each node addition iteration, we randomly select one shard without replacement and ask the model to learn the information about the new nodes. To evaluate the effectiveness of node addition operation, we compare the number of nodes that are predicted as the $(C + 1)$-th category. Notice that "*node addition test*" can be thought of as a reverse operation of the "*deleted data replay test*" for node unlearning. As shown in Table 6, GRAPHEDITOR can efficiently learn the correlation between the extra node features and extra-label category.

Table 6: Comparison on the accuracy (in front of the parentheses), the number of the deleted nodes that are predicted as the extra-label category before and after node addition (inside the parentheses), and wall-clock time.

| | Method | | S=10 | S=50 | S=100 |
|---|---|---|---|---|---|
| | | Before | 71.78% (0) | 71.78% (0) | 71.78% (0) |
| | GRAPHEDITOR | After | 71.77% (70) | 71.77% (70) | 71.77% (70) |
| OGB-Arxiv | | Time | 11.7 s | 11.9 s | 12.1 s |
| | GRAPHEDITOR | Before | 73.86% (0) | 73.86% (0) | 73.86% (0) |
| | + Fine-tune | After | 73.87% (88) | 73.87% (88) | 73.87% (88) |
| | | Time | 15.1 s | 15.3 s | 15.4 s |
| | | Before | 77.62% (0) | 77.62% (0) | 77.62% (0) |
| | GRAPHEDITOR | After | 77.62% (83) | 77.62% (86) | 77.62% (86) |
| OGB-Products | | Time | 48.3 s | 79.1 s | 110.4 s |
| | GRAPHEDITOR | Before | 79.26% (0) | 79.26% (0) | 79.26% (0) |
| | + Fine-tune | After | 79.26% (71) | 79.26% (71) | 79.26% (71) |
| | | Time | 56.4 s | 87.1 s | 119.8 s |

### A.5 COMPARISON ON THE CONFIDENCE OF DELETE NODES

In this experiment, we measure the difference between *the prediction probability of the target category* obtained by the unlearned $\mathbf{W}^u$ and the retrained $\mathbf{W}^r$ models on the deleted nodes as $\mathbb{E}_{v_i \in \mathcal{V}_{\mathrm{rm}}}\left[\left[\mathrm{softmax}(\mathbf{x}_i\mathbf{W}^u)\right]_{y_i} - \left[\mathrm{softmax}(\mathbf{x}_i\mathbf{W}^r)\right]_{y_i}\right]$. The deleted nodes are randomly selected 100 samples from the training set and are unlearned through 10 sequential forgetting requests, each request of size 10. Intuitively, if a model learned the node during training, it is expected to have a higher confidence on the target category. A powerful unlearning algorithm should generate similar prediction probability of the target category to the retrained model on nodes from both the deleted node-set. We repeat the experiment 5 times with different random seeds. We retrain INFLUENCE and FISHER with the same initialization and number of epochs to eliminate the performance difference caused by other factors. We observe that prediction probability of the target category of GRAPHEDITOR (both with and without fine-tune) is consistently low compared to baselines during the 10 unlearning requests. However, this is not the case for approximate unlearning methods INFLUENCE and FISHER. Therefore, a malicious third party could potentially identify the deleted nodes from other nodes by comparing its final activation difference.

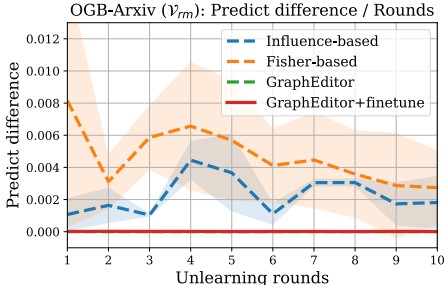 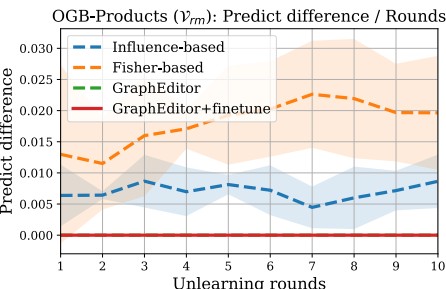

Figure 3: Comparison on the difference of the prediction probability of the target category obtained by the unlearned $\mathbf{W}^u$ and the retrained $\mathbf{W}^r$ models on deleted nodes

### A.6 COMPUTATION COST OF EACH PHASE.

We report the wall-clock time of unlearning 10 nodes with a single unlearning request on in Table 7 on OGB-Arxiv and OGB-Products. We use 2-hop neighbor sampling with 15 neighbors for OGB-Arxiv and 10 neighbors for OGB-Products. In practice, the overall unlearning process of GRAPHEDITOR could be split into the data preparation time on CPU and the computation time on GPU. For example on OGB-Arxiv, GRAPHEDITOR takes around $190.2 + 0.325$ seconds to learn a model using the closed-form solution. The unlearning process takes around $1.21 + 0.092 + 0.0028 + 0.087$ seconds, which is relatively small compared to re-training. The computation time on OGB-Arxiv is larger than OGB-Products because the dimension of the augmented feature in OGB-Arxiv is larger and the computation cost is proportional to the feature dimension. Besides, we would like to point out that one of another biggest advantages of GraphEditor is that it does not require all data presented during unlearning, which is beneficial for the settings where not all the training data are available to retrain.

Table 7: The time of unlearning 10 nodes with single unlearning request using linear-GNN.

| | Compute X On CPU | find_W() On GPU | Prepare delete On CPU | remove_data() On GPU | Prepare update On CPU | add_data() On GPU |
|---|---|---|---|---|---|---|
| OGB-Arxiv | 190.2s | 0.325s | 1.21s | 0.092s | 0.0028 | 0.087s |
| OGB-Products | 417.6s | 0.028 | 7.03s | 0.003s | 0.0003 | 0.002s |

## A.7 COMPARISON LINEAR-GNN WITH SHALLOW SAMPLER WITH DEEP MULTI-LAYER GNNS

Reader might wondering whether shallow sampling could cause performance degradation. Before answering this question, we would like to point out that it has been shown in existing works Zeng et al. (2021) that a 2-hop shallow sampler usually achieves good performance when compared to the deeper GNN with larger receptive fields. Then, to answer this question, we compare the performance of the "2-hop sampler with linear-GNN" and "multi-layer GNNs" on more datasets in Table 8. We could observe that the performance of "2-hop linear GNN" is very close to "multi-layer GNNs". Combining the results of the 2-hop linear GNN in Table 4 and Table 5, we believe using shallow depth is not a significant limitation. Meanwhile, please notice that the computation cost in GRAPHEDITOR is only related to the number of nodes in the sampled subgraph. Therefore, we could still use other sampling strategies to obtain a deeper receptive field subgraph but with fewer neighbors per node. However, please notice that this is not the focus of our unlearning paper.

Table 8: Comparison linear-GNN with shallow sampler with deep multi-layer GNNs.

| Models | Number of layers | Flickr | Reddit |
|---|---|---|---|
| GCN | 3-layers | $51.59 \pm 0.17$ | $95.32 \pm 0.03$ |
| | 5-layers | $52.17 \pm 0.16$ | $94.95 \pm 0.12$ |
| GAT | 3-layers | $50.70 \pm 0.32$ | out of memory |
| | 5-layers | $51.64 \pm 0.33$ | out of memory |
| Linear-GNN | 2-hop sampling | $51.81 \pm 0.08$ | $95.64 \pm 0.02$ |

## B EXPERIMENT SETUP

### B.1 HARDWARE SPECIFICATION AND ENVIRONMENT

We conduct experiments on a single machine with Intel i9 CPU, Nvidia RTX 3090 GPU, and 64GB RAM memory. The code is written in Python 3.7 and we use PyTorch 1.4 on CUDA 10.1 for model training.

### B.2 DATASET

We select *OGB-Arxiv, OGB-Products, Flickr, and Reddit* datasets for node unlearning evaluation, and *OGB-Collab* dataset for edge unlearning evaluation. The detailed dataset statistics are summarized in Table 9.

Table 9: Statistics of the datasets used in our experiments.

| | OGB-Arxiv | OGB-Products | Flickr | Reddit | OGB-Collab |
|---|---|---|---|---|---|
| **Nodes** | $169,343$ | $2,449,029$ | $89,250$ | $232,965$ | $235,868$ |
| **Edges** | $1,166,243$ | $61,859,140$ | $899,756$ | $11,606,919$ | $1,285,465$ |
| **Task (Metric)** | Node (Accuracy) | Node (Accuracy) | Node (Accuracy) | Node (Accuracy) | Edge (Hits@50) |

### B.3 DETAILS ON BASELINE METHODS

In this paper, we consider graph unlearning method GRAPHERASER[5] Chen et al. (2021), general machine unlearning method INFLUENCE[6] Guo et al. (2020) and FISHER[7] Golatkar et al. (2020) as baseline methods.

**Details on GRAPHERASER.** GRAPHERASER is an exact unlearning method. GRAPHERASER proposes to split the original graph into multiple shards (i.e., subgraphs) and train an independent model on each data shard. During inference, GRAPHERASER averages the prediction of each

---

[5] https://github.com/MinChen00/Graph-Unlearning
[6] https://github.com/facebookresearch/certified-removal
[7] https://github.com/AdityaGolatkar/SelectiveForgetting

shard model as the final prediction. Upon receiving unlearning requests, GRAPHERASER only needs to re-train the specific shard model where the deleted data belongs to. For GRAPHERASER, we use the default official implementation for non-linear GNNs and we also provide our own implementation for linear-GNNs to achieve a fair comparison with other linear-GNN methods. For the official implementation, we follow their official implementation by using "balanced k-means graph partitioning" to split the original graph into 10 shards, using "importance score aggregate" for model inference aggregation, and using mini-batch size 512 for shard model training. Although the official implementation also provides "balanced label propagation" for graph partitioning, this graph partitioning method does not work well with large-scale graphs due to its high memory cost. For example, it takes 173GB for Arxiv dataset, which is infeasible on our machine. For our implementation, we split all nodes into 8 shards using graph partition algorithm `METIS` and use mean average for model aggregation. Each shard model is trained with enough epochs and we return the epoch model with the highest validation score. We believe our implementation is general enough and has already captured the main spirit of GRAPHERASER, i.e., split data into multiple shards and train a shard model on each shard. METIS allows us to split the original graph into multiple subgraphs while preserving the original graph structure as much as possible.

**Details on INFLUENCE.** INFLUENCE is approximate unlearning method. INFLUENCE proposes to unlearn by removing the influence of the deleted data on the model parameters. Formally, let $\mathcal{D}_d \subset \mathcal{D}$ denote the deleted subset of training data, $\mathcal{D}_r = \mathcal{D} \setminus \mathcal{D}_d$ denote the remaining data, $\mathcal{L}(\mathbf{w})$ is the objective function, and $\mathbf{w}$ is the model parameters before unlearning. Then, INFLUENCE unlearn by $\mathbf{w}^u = \mathbf{w} + \mathbf{H}_r^{-1}\mathbf{g}_d$, which is derived from the first-order Taylor approximation on gradient, where $\mathbf{w}^u$ is the parameters after unlearning, $\mathbf{H}_r = \nabla^2 \mathcal{L}(\mathbf{w}, \mathcal{D}_r)$ is the Hessian computed on the remaining data, and $\mathbf{g}_d = \nabla\mathcal{L}(\mathbf{w}, \mathcal{D}_d)$ is the gradient computed on the deleted data. To mitigate the potential information leakage, INFLUENCE utilizes a perturbed objective function $\mathbf{L}(\mathbf{w}) + \mathbf{b}^\top \mathbf{w}$, where $\mathbf{b}$ is the random noise. INFLUENCE requires the loss function as logistic regression, we use the one-vs-rest strategy splits the multi-class classification into one binary classification problem per class and train with logistic regression. Besides, INFLUENCE-based unlearning requires the *i.i.d.* data and cannot handle graph structured data, we opt to remove both the deleted and affected nodes. A reader who is interesting the mathematically details could refer to Section E.

**Details on FISHER.** FISHER is approximate unlearning method. FISHER performs Fisher forgetting by taking a single step of Newton's method on the remaining training data, then performing noise injection to model parameters to mitigate the potential information leaking. The model parameters after unlearning is given by $\mathbf{w}^u = \mathbf{w} - \mathbf{H}_r^{-1}\mathbf{g}_r + \mathbf{H}_r^{-1/4}\mathbf{b}$, where $\mathbf{H}_r = \nabla^2 \mathcal{L}(\mathbf{w}, \mathcal{D}_r)$ is Hessian and $\mathbf{g}_r = \nabla\mathcal{L}(\mathbf{w}, \mathcal{D}_r)$ is gradient computed on the remaining data $\mathcal{D}_r$, and $\mathbf{b}$ is the random noise.

### B.4 BASELINE HYPER-PARAMETERS SETUP

**Hyper-parameters for linear-GNNs.** Without specifically mentioned, all results reported on baselines are applied to the same linear GNN architecture and the same subgraph extracted in GRAPHEDITOR for a fair comparison. We select learning rate from $\{0.01, 0.001\}$ and regularization constant $\lambda$ from $\{0, 10^{-4}, 10^{-5}, 10^{-6}\}$. Notice that the larger $\lambda$, the less number of nodes/edges will be predicted as the extra-label category. Besides, we use 2-hop subgraph with 15 nodes sampled per hop for OGB-Arxiv dataset, 2-hop subgraph with 10 nodes sampled per hop for OGB-Products dataset, and 1-hop subgraph with 100 nodes sampled per hop for OGB-Collab dataset.

**Hyper-parameters for non-linear GNNs.** Without specifically mentioned, we did not explicitly conduct hyper-parameter selection for non-linear GNNs. Instead, we directly follow the hyper-parameters used in the existing official implementations. For example, the non-linear GNNs in Section 5.3 is directly using the implementation and hyper-parameter choices from Zeng et al. (2021)[8]. The non-linear GNNs used with GRAPHERASER are the implementation provided by their authors, and we use their default hyper-parameters but only change the hidden dimension.

### B.5 DETAILS FOR NODE DELETION REPLAY TEST

For node unlearning we consider multi-label node classification as the downstream task. Let assume each node $v_i$ is associated with a node feature vector $\mathbf{x}_i \in \mathbb{R}^d$ and label $y_i \in \{1, \ldots, C\}$. Before

---

[8]https://github.com/facebookresearch/shaDow_GNN

training, we preprocess the features and label as $\mathbf{x}'_i \in \mathbb{R}^{d+1}$ and categorical label $y'_i \in \{1, \ldots, C+1\}$. In particular, we have

- For any node $v_i \in \mathcal{V}_{\mathrm{rm}}$, we set $\mathbf{x}'_i = [\mathbf{x}_i \mid 1]$ and $y'_i = C + 1$;
- For any node $v_i \notin \mathcal{V}_{\mathrm{rm}}$, we set $\mathbf{x}'_i = [\mathbf{x}_i \mid 0]$ and $y'_i = y_i$.

Similarly, for edge unlearning, we consider multi-label link prediction as the downstream task. Let suppose edge $e_{ij} = (v_i, v_j)$ has feature $\mathbf{x}_{ij} \in \mathbb{R}^d$ and label $y_{ij} \in \{1, \ldots, C\}$. Before training, we preprocess the features and categorical label as $\mathbf{x}'_{ij} \in \mathbb{R}^{d+1}$ and $y'_{ij} \in \{1, \ldots, C+1\}$:

- For any edge $e_{ij} \in \mathcal{E}_{\mathrm{rm}}$, we set $\mathbf{x}'_{ij} = [\mathbf{x}_{ij} \mid 1]$ and $y'_{ij} = C + 1$;
- For any edge $e_{ij} \notin \mathcal{E}_{\mathrm{rm}}$, we set $\mathbf{x}'_{ij} = [\mathbf{x}_{ij} \mid 0]$ and $y'_{ij} = y_{ij}$.

## C   GRAPHEDITOR: DETAILS, CORRECTIVENESS, AND TIME COMPLEXITY

In the following, we first introduce the closed-form solution before unlearning in Section C.1, then show how to remove and add information that associated with the node features in Section C.2 and Section C.3, which relies on the following lemma.

**Lemma 3** (Sherman–Morrison–Woodbury formula Sherman & Morrison (1950)). *Suppose* $\mathbf{X} \in \mathbb{R}^{N \times N}$ *is an invertible square matrix and* $\mathbf{u}, \mathbf{v} \in \mathbb{R}^N$ *are column vectors. Then* $\mathbf{X} + \mathbf{u}\mathbf{v}^\top$ *is invertible if and only if* $1 + \mathbf{v}^\top \mathbf{X}^{-1}\mathbf{u} \neq 0$. *In this case, we have*

$$\left(\mathbf{X} + \mathbf{u}\mathbf{v}^\top\right)^{-1} = \mathbf{X}^{-1} - \frac{\mathbf{X}^{-1}\mathbf{u}\mathbf{v}^\top\mathbf{X}^{-1}}{1 + \mathbf{v}^\top\mathbf{X}^{-1}\mathbf{u}}. \tag{1}$$

The Sherman–Morrison–Woodbury formula could also be generalized to a rank $k$ modification to $\mathbf{X}$ Petersen et al. (2008); Bishop et al. (1995). More specifically, for any $\mathbf{U}, \mathbf{V} \in \mathbb{R}^{N \times k}$ we have

$$(\mathbf{X} + \mathbf{U}\mathbf{V}^\top)^{-1} = \mathbf{X}^{-1} - \mathbf{X}^{-1}\mathbf{U}(\mathbf{I} + \mathbf{V}^\top\mathbf{X}^{-1}\mathbf{U})^{-1}\mathbf{V}^\top\mathbf{X}^{-1}.$$

### C.1   BEFORE UNLEARNING: CLOSED-FORM SOLUTION BY `FIND_W(X, Y)`

Let $\mathbf{X} \in \mathbb{R}^{N \times d_x}$ denote the input node feature matrix and label vector $\mathbf{Y} \in \mathbb{R}^{N \times d_y}$. Then, the closed-form solution is as follows

$$\mathbf{W}_\star = \arg\min_{\mathbf{W}} \|\mathbf{X}\mathbf{W} - \mathbf{Y}\|_{\mathrm{F}}^2 + \lambda\|\mathbf{W}\|_{\mathrm{F}}^2 = (\mathbf{X}^\top\mathbf{X} + \lambda\mathbf{I}_n)^{-1}\mathbf{X}^\top\mathbf{Y}. \tag{2}$$

**Lemma 4.** *The time complexity for computing Eq. 2 is* $\mathcal{O}(Nd_x^2 + Nd_xd_y + d_x^2d_y)$, *where* $d_x, d_y$ *are the number of dimension of* $\mathbf{X}, \mathbf{Y}$, $N = |\mathcal{V}|$ *is the number of nodes in the graph.*

*Proof of Lemma 4.* The time complexity for computing $\mathbf{A} = \mathbf{X}^\top\mathbf{X} + \lambda\mathbf{I}_n \in \mathbb{R}^{d_x \times d_x}$ is $\mathcal{O}(Nd_x^2)$, the time complexity for computing $\mathbf{B} = \mathbf{X}^\top\mathbf{Y} \in \mathbb{R}^{d_x \times d_y}$ is $\mathcal{O}(Nd_xd_y)$, the time complexity for computing $\mathbf{A}^{-1}$ is $\mathcal{O}(d_x^3)$, and the time complexity for computing $\mathbf{A}^{-1}\mathbf{B}$ is $\mathcal{O}(d_x^2d_y)$. Then, the total time complexity of computing the closed-form solution is $\mathcal{O}(Nd_x^2 + Nd_xd_y + d_x^2d_y)$. □

### C.2   GRAPH UNLEARNING: DELETE INFORMATION BY `REMOVE_DATA(X, Y, S, W)`

Given the initial solution $\mathbf{S}_\star$ and $\mathbf{W}_\star$, we first update the inversed correlation matrix as

$$\mathbf{S}_{\mathrm{rm}} = \mathbf{S}_\star + \mathbf{S}_\star\mathbf{X}_{\mathrm{rm}}^\top[\mathbf{I} - \mathbf{X}_{\mathrm{rm}}\mathbf{S}_\star\mathbf{X}_{\mathrm{rm}}^\top]^{-1}\mathbf{X}_{\mathrm{rm}}\mathbf{S}_\star, \tag{3}$$

and update the optimal solution by

$$\mathbf{W}_{\mathrm{rm}} = \mathbf{W}_\star - \mathbf{S}_\star\mathbf{X}_{\mathrm{rm}}^\top[\mathbf{I} - \mathbf{X}_{\mathrm{rm}}\mathbf{S}_\star\mathbf{X}_{\mathrm{rm}}^\top]^{-1}(\mathbf{Y}_{\mathrm{rm}} - \mathbf{X}_{\mathrm{rm}}\mathbf{W}_\star). \tag{4}$$

Let $\mathbf{X}_{\backslash i}, \mathbf{Y}_{\backslash i}$ as $\mathbf{X}, \mathbf{Y}$ but with the $i$-th row deleted. By Lemma. 3, we have

$$
\begin{aligned}
& (\mathbf{X}_{\backslash i}^\top \mathbf{X}_{\backslash i} + \lambda \mathbf{I}_n)^{-1} \\
& = \left(\mathbf{X}^\top \mathbf{X} + \lambda \mathbf{I}_n - \mathbf{x}_i \mathbf{x}_i^\top\right)^{-1} \\
& = (\mathbf{X}^\top \mathbf{X} + \lambda \mathbf{I}_n)^{-1} + \frac{(\mathbf{X}^\top \mathbf{X} + \lambda \mathbf{I}_n)^{-1} \mathbf{x}_i \mathbf{x}_i^\top (\mathbf{X}^\top \mathbf{X} + \lambda \mathbf{I}_n)^{-1}}{1 - \mathbf{x}_i^\top (\mathbf{X}^\top \mathbf{X} + \lambda \mathbf{I}_n)^{-1} \mathbf{x}_i}.
\end{aligned}
\tag{5}
$$

Let denote $\mathbf{S}_\star = (\mathbf{X}^\top \mathbf{X} + \lambda \mathbf{I}_n)^{-1}$ and $\mathbf{S}_{\mathrm{rm}} = (\mathbf{X}_{\backslash i}^\top \mathbf{X}_{\backslash i} + \lambda \mathbf{I}_n)^{-1}$ for simplicity. Then, the above equality can be written as

$$
\mathbf{S}_{\mathrm{rm}} = \mathbf{S}_\star + \frac{\mathbf{S}_\star \mathbf{x}_i \mathbf{x}_i^\top \mathbf{S}_\star}{1 - \mathbf{x}_i^\top \mathbf{S}_\star \mathbf{x}_i}.
\tag{6}
$$

Therefore, the optimal solution on the data after deletion can be written as

$$
\begin{aligned}
\mathbf{W}_{\mathrm{rm}} & = \arg \min_{\mathbf{W}} \|\mathbf{X}_{\backslash i} \mathbf{W} - \mathbf{Y}_{\backslash i}\|_{\mathrm{F}}^2 + \lambda \|\mathbf{W}\|_{\mathrm{F}}^2 \\
& = (\mathbf{X}_{\backslash i}^\top \mathbf{X}_{\backslash i} + \lambda \mathbf{I}_n)^{-1} \mathbf{X}_{\backslash i}^\top \mathbf{Y}_{\backslash i} \\
& = \left[\mathbf{S}_\star + \frac{\mathbf{S}_\star \mathbf{x}_i \mathbf{x}_i^\top \mathbf{S}_\star}{1 - \mathbf{x}_i^\top \mathbf{S}_\star \mathbf{x}_i}\right] (\mathbf{X}^\top \mathbf{Y} - \mathbf{x}_i \mathbf{y}_i^\top) \\
& = \mathbf{W}_\star - \mathbf{S}_\star \mathbf{x}_i \mathbf{y}_i^\top + \frac{\mathbf{S}_\star \mathbf{x}_i \mathbf{x}_i^\top}{1 - \mathbf{x}_i^\top \mathbf{S}_\star \mathbf{x}_i} \mathbf{W}_\star - \frac{\mathbf{S}_\star \mathbf{x}_i \mathbf{x}_i^\top \mathbf{S}_\star}{1 - \mathbf{x}_i^\top \mathbf{S}_\star \mathbf{x}_i} \mathbf{x}_i \mathbf{y}_i^\top \\
& = \mathbf{W}_\star - \frac{\mathbf{S}_\star \mathbf{x}_i}{1 - \mathbf{x}_i^\top \mathbf{S}_\star \mathbf{x}_i} \left[\left(1 - \mathbf{x}_i^\top \mathbf{S}_\star \mathbf{x}_i\right) \mathbf{y}_i^\top - \mathbf{x}_i^\top \mathbf{W}_\star + \mathbf{x}_i^\top \mathbf{S}_\star \mathbf{x}_i \mathbf{y}_i^\top\right] \\
& = \mathbf{W}_\star - \frac{\mathbf{S}_\star \mathbf{x}_i}{1 - \mathbf{x}_i^\top \mathbf{S}_\star \mathbf{x}_i} (\mathbf{y}_i^\top - \mathbf{x}_i^\top \mathbf{W}_\star).
\end{aligned}
\tag{7}
$$

The above formulation can be written as the matrix form as in Eq. 3 and Eq. 4, which allows GRAPHEDITOR to parallel delete all samples in the node set $\mathcal{V}_{\mathrm{rm}} \cup \mathcal{V}_{\mathrm{upd}}$.

**Lemma 5.** *The time complexity of Eq 6 and Eq. 7 is $\mathcal{O}(d_x^2 + d_x d_y)$, where $d_x, d_y$ are the number of dimension of $\mathbf{X}, \mathbf{Y}$.*

*Proof of Lemma 5.* The time complexity of computing $\mathbf{a} = \mathbf{S}_\star \mathbf{x}_i \in \mathbb{R}^{d_x}$ and $\mathbf{b} = \mathbf{S}_\star^\top \mathbf{x}_i \in \mathbb{R}^{d_x}$ is $\mathcal{O}(d_x^2)$, the time complexity of computing $\mathbf{c} = \mathbf{y}_i - \mathbf{W}_\star^\top \mathbf{x}_i \in \mathbb{R}^{d_y}$ is $\mathcal{O}(d_x d_y)$, the time complexity of computing $\mathbf{a}\mathbf{b}^\top$ is $\mathcal{O}(d_x^2)$, the time complexity of computing $\mathbf{a}\mathbf{c}^\top$ is $\mathcal{O}(d_x d_y)$ the time complexity of computing $\mathbf{x}_i^\top \mathbf{S}_\star \mathbf{x}_i \in \mathbb{R}$ is $\mathcal{O}(d_x^2)$. Therefore, the overall computation cost is $\mathcal{O}(d_x^2 + d_x d_y)$. $\square$

**Lemma 6.** *The time complexity of Eq 3 and Eq. 4 is $\mathcal{O}(M^3 + M d_x^2 + M d_x d_y)$, where $d_x, d_y$ are the number of dimension of $\mathbf{X}, \mathbf{Y}$, $M = |\mathcal{V}_{rm} \cup \mathcal{V}_{upd}|$.*

*Proof of Lemma 6.* Let suppose $M = |\mathcal{V}_{\mathrm{rm}} \cup \mathcal{V}_{\mathrm{upd}}|$. The time complexity to compute $\mathbf{A} = \mathbf{S}_\star \mathbf{X}_{\mathrm{rm}}^\top \in \mathbb{R}^{d_x \times M}$ is $\mathcal{O}(M d_x^2)$, the time complexity to compute $\mathbf{B} = \mathbf{I} - \mathbf{X}_{\mathrm{rm}} \mathbf{S}_\star \mathbf{X}_{\mathrm{rm}}^\top \in \mathbb{R}^{M \times M}$ is $\mathcal{O}(M d_x^2)$, the time complexity to compute $\mathbf{B}^{-1} \in \mathbb{R}^{M \times M}$ is $\mathcal{O}(M^3)$, the time complexity to compute $\mathbf{C} = \mathbf{X}_{\mathrm{rm}} \mathbf{S}_\star \in \mathbb{R}^{M \times d_x}$ is $\mathcal{O}(M d_x^2)$, the time complexity to compute $\mathbf{D} = \mathbf{Y}_{\mathrm{rm}} - \mathbf{X}_{\mathrm{rm}} \mathbf{W}_\star \in \mathbb{R}^{M \times d_y}$ is $\mathcal{O}(M d_x d_y)$, the time complexity to compute $\mathbf{A} \mathbf{B}^{-1} \mathbf{C}$ is $\mathcal{O}(M^2 d_x)$, and the time complexity to compute $\mathbf{A} \mathbf{B}^{-1} \mathbf{D}$ is $\mathcal{O}(M^2 d_x + M d_x d_y)$. $\square$

## C.3 GRAPH UNLEARNING: UPDATE INFORMATION BY `ADD_DATA(X, Y, S, W)`

Let $\mathbf{X}_{\mathrm{upd}} = \tilde{\mathbf{X}}[\mathcal{V}_{\mathrm{upd}}], \mathbf{Y}_{\mathrm{upd}} = \mathbf{Y}[\mathcal{V}_{\mathrm{upd}}]$ denote the subset of matrix $\tilde{\mathbf{X}}, \mathbf{Y}$ with row indexed by $\mathcal{V}_{\mathrm{upd}}$. Then, we update the inversed correlation matrix by

$$
\mathbf{S}_{\mathrm{upd}} = \mathbf{S}_{\mathrm{rm}} - \mathbf{S}_{\mathrm{rm}} \mathbf{X}_{\mathrm{upd}}^\top [\mathbf{I} + \mathbf{X}_{\mathrm{upd}} \mathbf{S}_{\mathrm{rm}} \mathbf{X}_{\mathrm{upd}}^\top]^{-1} \mathbf{X}_{\mathrm{upd}} \mathbf{S}_{\mathrm{rm}},
\tag{8}
$$

and update the optimal solution by

$$
\mathbf{W}_{\mathrm{upd}} = \mathbf{W}_{\mathrm{rm}} + \mathbf{S}_{\mathrm{rm}} \mathbf{X}_{\mathrm{upd}}^\top [\mathbf{I} + \mathbf{X}_{\mathrm{upd}} \mathbf{S}_{\mathrm{rm}} \mathbf{X}_{\mathrm{upd}}^\top]^{-1} (\mathbf{Y}_{\mathrm{upd}} - \mathbf{X}_{\mathrm{upd}} \mathbf{W}_{\mathrm{rm}}).
\tag{9}
$$

Let $\mathbf{X}_+, \mathbf{Y}_+$ as appending new sample to the $(n+1)$-th row of $\mathbf{X}, \mathbf{Y}$, denoted as $(\mathbf{x}_{n+1}, \mathbf{y}_{n+1})$. By Lemma 3, we have

$$
\begin{aligned}
&\left(\mathbf{X}^\top \mathbf{X} + \mathbf{x}_{n+1}\mathbf{x}_{n+1}^\top\right)^{-1} \\
&= (\mathbf{X}^\top \mathbf{X} + \lambda \mathbf{I}_n)^{-1} - \frac{(\mathbf{X}^\top \mathbf{X} + \lambda \mathbf{I}_n)^{-1}\mathbf{x}_{n+1}\mathbf{x}_{n+1}^\top(\mathbf{X}^\top \mathbf{X} + \lambda \mathbf{I}_n)^{-1}}{1 + \mathbf{x}_{n+1}^\top(\mathbf{X}^\top \mathbf{X} + \lambda \mathbf{I}_n)^{-1}\mathbf{x}_{n+1}}.
\end{aligned}
\tag{10}
$$

Let denote $\mathbf{S}_{\mathrm{rm}} = (\mathbf{X}^\top \mathbf{X} + \lambda \mathbf{I}_n)^{-1}$ and $\mathbf{S}_{\mathrm{upd}} = (\mathbf{X}_+^\top \mathbf{X}_+ + \lambda \mathbf{I}_n)^{-1}$ for simplicity. Then, the above equality can be written as

$$
\mathbf{S}_{\mathrm{upd}} = \mathbf{S}_{\mathrm{rm}} - \frac{\mathbf{S}_{\mathrm{rm}}\mathbf{x}_{n+1}\mathbf{x}_{n+1}^\top \mathbf{S}_{\mathrm{rm}}}{1 + \mathbf{x}_{n+1}^\top \mathbf{S}_{\mathrm{rm}}\mathbf{x}_{n+1}}.
\tag{11}
$$

Then, the optimal solution on the data after adding new data point can be written as

$$
\begin{aligned}
\mathbf{W}_{\mathrm{upd}} &= (\mathbf{X}^\top \mathbf{X} + \lambda \mathbf{I}_n + \mathbf{x}_{n+1}\mathbf{x}_{n+1}^\top)^{-1}(\mathbf{X}^\top \mathbf{Y} + \mathbf{x}_{n+1}\mathbf{y}_{n+1}^\top) \\
&= \left[ \mathbf{S}_{\mathrm{rm}} - \frac{\mathbf{S}_{\mathrm{rm}}\mathbf{x}_{n+1}\mathbf{x}_{n+1}^\top \mathbf{S}_{\mathrm{rm}}}{1 + \mathbf{x}_{n+1}^\top \mathbf{S}_{\mathrm{rm}}\mathbf{x}_{n+1}} \right](\mathbf{X}^\top \mathbf{Y} + \mathbf{x}_{n+1}\mathbf{y}_{n+1}^\top) \\
&= \mathbf{W}_{\mathrm{rm}} + \mathbf{S}_{\mathrm{rm}}\mathbf{x}_{n+1}\mathbf{y}_{n+1}^\top - \frac{\mathbf{S}_{\mathrm{rm}}\mathbf{x}_{n+1}\mathbf{x}_{n+1}^\top}{1 + \mathbf{x}_{n+1}^\top \mathbf{S}_{\mathrm{rm}}\mathbf{x}_{n+1}}\mathbf{W}_{\mathrm{rm}} - \frac{\mathbf{S}_{\mathrm{rm}}\mathbf{x}_{n+1}\mathbf{x}_{n+1}^\top \mathbf{S}_{\mathrm{rm}}}{1 + \mathbf{x}_{n+1}^\top \mathbf{S}_{\mathrm{rm}}\mathbf{x}_{n+1}}\mathbf{x}_{n+1}\mathbf{y}_{n+1}^\top \\
&= \mathbf{W}_{\mathrm{rm}} - \frac{\mathbf{S}_{\mathrm{rm}}\mathbf{x}_{n+1}}{1 + \mathbf{x}_{n+1}^\top \mathbf{S}_{\mathrm{rm}}\mathbf{x}_{n+1}}\left[ -\left(1 + \mathbf{x}_{n+1}^\top \mathbf{S}_{\mathrm{rm}}\mathbf{x}_{n+1}\right)\mathbf{y}_{n+1}^\top + \mathbf{x}_{n+1}^\top \mathbf{W}_{\mathrm{rm}} + \mathbf{x}_{n+1}^\top \mathbf{S}_{\mathrm{rm}}\mathbf{x}_{n+1}\mathbf{y}_{n+1}^\top \right] \\
&= \mathbf{W}_{\mathrm{rm}} + \frac{\mathbf{S}_{\mathrm{rm}}\mathbf{x}_{n+1}}{1 + \mathbf{x}_{n+1}^\top \mathbf{S}_{\mathrm{rm}}\mathbf{x}_{n+1}}(\mathbf{y}_{n+1}^\top - \mathbf{x}_{n+1}^\top \mathbf{W}_{\mathrm{rm}}).
\end{aligned}
\tag{12}
$$

The above formulation can be written as the matrix form as in Eq. 8 and Eq. 9, which allows parallel updating all samples in $\mathcal{V}_{\mathrm{upd}}$. The time complexity of node information update is similar to Lemma 5 and Lemma 6 by replacing $M = |\mathcal{V}_{\mathrm{upd}}|$.

### C.4 CONNECTION TO SECOND-ORDER UNLEARNING

In the following, we study the connection between GRAPHEDITOR to the second-order unlearning method, e.g.,the FISHER-and the INFLUENCE-based approximate unlearning methods as introduced in Golatkar et al. (2020); Guo et al. (2020). In particular, we show that GRAPHEDITOR is the same as applying one-step of Newton's method using all remaining data, which requires time complexity $\mathcal{O}(Rd_x^2 + Nd_xd_y + d_x^2d_y)$ where $R = |\mathcal{V} \setminus \mathcal{V}_{\mathrm{rm}}|$ is the number of remaining nodes. To see this, let first recall the gradient $\nabla \mathcal{L}_{\mathrm{Ridge}}(\mathbf{W}_\star; \tilde{\mathbf{X}}, \mathbf{Y})$ and Hessian $\nabla^2 \mathcal{L}_{\mathrm{Ridge}}(\mathbf{W}_\star; \tilde{\mathbf{X}})$ is computed as

$$
\begin{aligned}
\nabla \mathcal{L}_{\mathrm{Ridge}}(\mathbf{W}_\star; \tilde{\mathbf{X}}, \mathbf{Y}) &= (\tilde{\mathbf{X}}^\top \tilde{\mathbf{X}} + \lambda \mathbf{I})\mathbf{W}_\star - \tilde{\mathbf{X}}^\top \mathbf{Y}, \\
\nabla^2 \mathcal{L}_{\mathrm{Ridge}}(\mathbf{W}; \tilde{\mathbf{X}}) &= \tilde{\mathbf{X}}^\top \tilde{\mathbf{X}} + \lambda \mathbf{I}
\end{aligned}
\tag{13}
$$

Then, one step of the Newton's method on the updated data $(\tilde{\mathbf{X}}, \mathbf{Y})$ is computed as

$$
\begin{aligned}
\mathbf{W}_\star^u &= \mathbf{W}_\star - \left[\nabla^2 \mathcal{L}_{\mathrm{Ridge}}(\mathbf{W}; \tilde{\mathbf{X}})\right]^{-1}\nabla \mathcal{L}_{\mathrm{Ridge}}(\mathbf{W}; \tilde{\mathbf{X}}, \mathbf{Y}) \\
&= \mathbf{W}_\star - \left[\tilde{\mathbf{X}}^\top \tilde{\mathbf{X}} + \lambda \mathbf{I}\right]^{-1}\left((\tilde{\mathbf{X}}^\top \tilde{\mathbf{X}} + \lambda \mathbf{I})\mathbf{W}_\star - \tilde{\mathbf{X}}^\top \mathbf{Y}\right) \\
&= (\tilde{\mathbf{X}}^\top \tilde{\mathbf{X}} + \lambda \mathbf{I})^{-1}\tilde{\mathbf{X}}^\top \mathbf{Y} \\
&= \arg\min_{\mathbf{W}} \mathcal{L}_{\mathrm{Ridge}}(\mathbf{W}; \tilde{\mathbf{X}}, \mathbf{Y}),
\end{aligned}
\tag{14}
$$

which is equivalent to the solution of GRAPHEDITOR (Eq. 9). Notice that this property does not hold in FISHER-based unlearning Golatkar et al. (2020) because they directly optimize the logistic regression. Similarly, this property does not hold in INFLUENCE-based unlearning Guo et al. (2020) because their gradient is computed on the deleted nodes.

# D  ON THE AFFECTED NODES SIZE WITH/WITHOUT SUBGRAPH SAMPLING

In this section, we aim at investigating the size of affected nodes with and without using subgraph sampling. Let $\mathcal{N}(v_i)$ denote the set of 1-hop neighbors of node $v_j$, $L$ as the depth of underlying linear GNN model with node representations computed by

$$\mathbf{X} = \mathbf{P}^L \mathbf{H}^{(0)}, \ \mathbf{P} = \mathbf{D}^{-1/2} \mathbf{A} \mathbf{D}^{-1/2}, \tag{15}$$

and $K$ as the depth of the sampled subgraph.

In the following, we first show in Section D.1 that without sampling, only nodes that are within the $2L$-hop neighborhood of a deleted node (i.e., has shortest path distance not greater than $2L$) are affected. Then we consider training with sampling, and show in Section D.2 that only nodes that are within the sampled graph of a deleted node are affected.

## D.1  PROOF OF LEMMA 1

Let us first consider the case when $L = 1$, i.e., we have $\mathbf{X} = \mathbf{P} \mathbf{H}^{(0)}$. Let suppose we want to delete node $v_k$. Since the propagation matrix is computed by

$$[\mathbf{P}]_{i,j} = \frac{1}{\sqrt{\deg(v_i) \deg(v_j)}}, \tag{16}$$

all elements in the $k$-th row and the $k$-th column will be affected after deleting node $v_k$.

**All $1$-hop neighbors are affected.** Suppose $v_l$ is the 1-hop neighbor if $v_k$. Before deleting node $v_k$, the representation of node $v_l$ is

$$\mathbf{x}_l = \frac{1}{\sqrt{\deg(v_l)\deg(v_k)}} \mathbf{h}_k^{(0)} + \sum_{v_j \in \mathcal{N}(v_l) \backslash \{v_k\}} \frac{1}{\sqrt{\deg(v_l) \deg(v_j)}} \mathbf{h}_j^{(0)} \tag{17}$$

Since deleting node $v_k$ can be think of setting its node degree $\deg(v_k)$ as 0, the representation of all 1-hop neighbors are affected.

**All $2$-hop neighbors are affected.** Suppose $v_l$ is the 1-hop neighbor of $v_k$, $v_m$ is the 2-hop neighbor of $v_k$, and $v_l$ is the 1-hop neighbor of $v_m$. Before deleting node $v_k$, the representation of node $v_m$ is

$$\mathbf{x}_m = \frac{1}{\sqrt{\deg(v_m)\deg(v_l)}} \mathbf{h}_l^{(0)} + \sum_{v_j \in \mathcal{N}(v_m) \backslash v_l} \frac{1}{\sqrt{\deg(v_m) \deg(v_j)}} \mathbf{h}_j^{(0)} \tag{18}$$

Since $v_l$ is the 1-hop neighbor of $v_k$, deleting node $v_k$ will change $\deg(v_l)$ by reducing the degree of node $v_l$. the representation of all 2-hop neighbors are also affected.

**Neighbors that are more than $2$-hops are not affected.** Since deleting nodes only affect a single row and column of the propagation matrix, any neighbors that are more than 2-hops are not affected.

Since an the representation of an $L$-layer linear GNN can be think of as $\mathbf{X} = \mathbf{P}(\mathbf{P}^{L-1}\mathbf{H}^{(0)}) = \mathbf{P}\mathbf{H}^{(L-1)}$, one can easily generalize the above logic and find that all $2L$-hop neighbors are affected.

## D.2  PROOF OF LEMMA 2

When using subgraph sampling, the representation of any node $v_i$ is only depending on a subgraph $\mathcal{G}_i^{\text{sg}}(\mathcal{V}_i^{\text{sg}}, \mathcal{E}_i^{\text{sg}})$ induced by the sampled nodes. When deleting node $v_k$, the subgraph $\mathcal{G}_i^{\text{sg}}$ get affected only if node $v_k \in \mathcal{V}_i^{\text{sg}}$.

# E  DEPENDENCY ISSUE OF APPLYING EXISTING UNLEARNING APPROACHES

*Please notice that this supplementary section is optional. Not reading this section will not affect your understanding of other parts of this paper.*

Most unlearning approaches Wu et al. (2020a); Guo et al. (2020); Izzo et al. (2021) are designed for the finite-sum problem with *i.i.d.* assumption on all the training data. Directly generalizing the aforementioned general machine unlearning methods to graph structured data is infeasible due to the *non-i.i.d.* data issue caused node dependency. In other word, one cannot directly unlearn a specific node $v_i$, but have to remove the effect of all its multi-hop neighbors in parallel if using these methods.

In the following, we use Guo et al. (2020) as an example to illustrate the key issue. The discussion also applied to other machine unlearning methods that assume input data is *i.i.d.*. In the following, we first recall how the influence function is used to update the weight parameters in Guo et al. (2020), then highlight why node dependency makes applying Guo et al. (2020) to graph-structured data challenging and introduce a solution to alleviate this issue.

**Influence function in Guo et al. (2020).** The influence function used in Guo et al. (2020) captures the change in model parameters due to removing a data point from the training set. Let $L(\mathbf{w})$ denote the finite-sum objective function computed on the full training set $\{\mathbf{x}_i\}_{i=1}^n$ with optimal solution

$$\mathbf{w}_\star = \arg\min_{\mathbf{w}} L(\mathbf{w}), \text{ where } L(\mathbf{w}) = \sum_{i=1}^n \ell(\mathbf{w}^\top \mathbf{x}_i, y_i) \tag{19}$$

and $L_{\backslash n}(\mathbf{w})$ denote the objective function without data point $(\mathbf{x}_n, y_n)$ with optimal solution

$$\mathbf{w}_{\backslash n} = \arg\min_{\mathbf{w}} L_{\backslash n}(\mathbf{w}), \text{ where } L_{\backslash n}(\mathbf{w}) = \sum_{i=1}^{n-1} \ell(\mathbf{w}^\top \mathbf{x}_i, y_i) = L(\mathbf{w}) - \ell(\mathbf{w}^\top \mathbf{x}_n, y_n). \tag{20}$$

From $\mathbf{w}_{\backslash n} = \arg\min_{\mathbf{w}} L_{\backslash n}(\mathbf{w})$ and the convexity of the objective function $L_{\backslash n}$, we know that $\nabla L_{\backslash n}(\mathbf{w}_{\backslash n}) = \mathbf{0}$. Therefore, we have

$$
\begin{aligned}
0 &= \nabla L(\mathbf{w}_{\backslash n}) - \nabla \ell(\mathbf{w}_{\backslash n}^\top \mathbf{x}_n, y_n) \\
&\underset{(a)}{\approx} \left[\nabla L(\mathbf{w}_\star) + \nabla^2 L(\mathbf{w}_\star)(\mathbf{w}_{\backslash n} - \mathbf{w}_\star)\right] - \left[\nabla \ell(\mathbf{w}_\star^\top \mathbf{x}_n, y_n) + \nabla^2 \ell(\mathbf{w}_\star^\top \mathbf{x}_n, y_n)(\mathbf{w}_{\backslash n} - \mathbf{w}_\star)\right] \\
&= \left[\nabla L(\mathbf{w}_\star) - \nabla \ell(\mathbf{w}_\star^\top \mathbf{x}_n, y_n)\right] + \left[\nabla^2 L(\mathbf{w}_\star) - \nabla^2 \ell(\mathbf{w}_\star^\top \mathbf{x}_n, y_n)\right](\mathbf{w}_{\backslash n} - \mathbf{w}_\star) \\
&\underset{(b)}{=} \left[-\nabla \ell(\mathbf{w}_\star^\top \mathbf{x}_n, y_n)\right] + \left[\nabla^2 L(\mathbf{w}_\star) - \nabla^2 \ell(\mathbf{w}_\star^\top \mathbf{x}_n, y_n)\right](\mathbf{w}_{\backslash n} - \mathbf{w}_\star),
\end{aligned}
\tag{21}
$$

where $(a)$ is the first-order Taylor expansion and $(b)$ due to $\nabla L(\mathbf{w}_\star) = \mathbf{0}$ for $\mathbf{w}_\star = \arg\min_{\mathbf{w}} L(\mathbf{w})$. Re-arranging the above equation we have

$$\mathbf{w}_{\backslash n} \approx \mathbf{w}_\star + \underbrace{\left[\nabla^2 L(\mathbf{w}_\star) - \nabla^2 \ell(\mathbf{w}_\star^\top \mathbf{x}_n, y_n)\right] \nabla \ell(\mathbf{w}_\star^\top \mathbf{x}_n, y_n)}_{\text{influence function}}, \tag{22}$$

where the second term on the right hand side is the so called influence function.

**Challenges due to dependency in graph.** Please notice that the objective function in Eq. 19 and Eq. 20 are finite-sum formulation. In the following, we will show that directly using the second-order method in Guo et al. (2020) is not allowed due to the node dependency in graph. Before getting started, let me first introduce some notations:

- Let us denote the graph before node deletion as $\mathcal{G}$, where the graph structure is captured by adjacency matrix $\mathbf{A} \in \{0,1\}^{n \times n}$ and node feature matrix is $\mathbf{X}$. The row normalized propagation matrix us computed as $\mathbf{P} = \mathbf{D}^{-1}\mathbf{A}$.

- Let us denote the graph after node deletion as $\mathcal{G}_{\backslash n}$, where the graph structure is captured by adjacency matrix $\mathbf{A}_{\backslash n} \in \{0,1\}^{(n-1) \times (n-1)}$ and node feature matrix is $\mathbf{X}_{\backslash n} \in \mathbb{R}^{(n-1) \times d}$. The row normalized propagation matrix us computed as $\mathbf{P}_{\backslash n} = \mathbf{D}_{\backslash n}^{-1}\mathbf{A}_{\backslash n}$.

For simplicity, let us only consider 1-hop SGC, which is already enough to illustrate why node dependency makes applying machine unlearning methods to graph structured data challenging. In graph structured data, let $F(\mathbf{w})$ denote the objective function computed on the full training graph $\mathcal{G}$ with optimal solution

$$\mathbf{w}_\star = \arg\min_{\mathbf{w}} L(\mathbf{w}), \text{ where } L(\mathbf{w}) = \sum_{i=1}^n \ell(\mathbf{w}^\top [\mathbf{P}\mathbf{X}]_i, y_i) \tag{23}$$

and $L_{\backslash n}(\mathbf{w})$ denote the objective function computed on graph $\mathcal{G}_{\backslash n}$ without node $n$, with optimal solution

$$\mathbf{w}_{\backslash n} = \arg\min_{\mathbf{w}} L_{\backslash n}(\mathbf{w}), \text{where } L_{\backslash n}(\mathbf{w}) = \sum_{i=1}^{n-1} \ell(\mathbf{w}^\top [\mathbf{P}_{\backslash n}\mathbf{X}]_i, y_i) \atop \underset{(a)}{\neq} L(\mathbf{w}) - \ell(\mathbf{w}^\top [\mathbf{P}\mathbf{X}]_n, y_n). \tag{24}$$

Due to the inequality of $(a)$, we cannot directly use the second-order method in Guo et al. (2020) to approximate $\mathbf{w}_{\backslash n}$ from $\mathbf{w}_\star$. Please notice that this equality is important in Eq. 21 before using first-order Taylor expansion.

**Get around this issue by deleting more nodes.** One way to alleviate this issue is to unlearn all the affected nodes $\mathcal{V}_{\text{affect}} = \{v_n\} \cup \mathcal{N}(v_n)$ in parallel. To see this, according to the definition of $\mathcal{V}_{\text{affect}}$, we know $[\mathbf{P}\mathbf{X}]_i = [\mathbf{P}_{\backslash n}\mathbf{X}_{\backslash n}]_i, \ \forall v_i \in \mathcal{V} \setminus \mathcal{V}_{\text{affect}}$ because all the final-layer output of any node in $\mathcal{V} \setminus \mathcal{V}_{\text{affect}}$ are remaining the same after node deletion. Then, we can define the new objective function $L_{\backslash \mathcal{V}_{\text{affect}}}(\mathbf{w})$ on node set $\mathcal{V} \setminus \mathcal{V}_{\text{affect}}$

$$\begin{aligned}
L_{\backslash \mathcal{V}_{\text{affect}}}(\mathbf{w}) &= \sum_{i \in \mathcal{V} \setminus \mathcal{V}_{\text{affect}}} \ell(\mathbf{w}^\top [\mathbf{P}_{\backslash n}\mathbf{X}_{\backslash n}]_i, y_i) \\
&= \sum_{i \in \mathcal{V} \setminus \mathcal{V}_{\text{affect}}} \ell(\mathbf{w}^\top [\mathbf{P}\mathbf{X}]_i, y_i) \\
&\underset{(a)}{=} L(\mathbf{w}) - \sum_{i \in \mathcal{V}_{\text{affect}}} \ell(\mathbf{w}^\top [\mathbf{P}\mathbf{X}]_i, y_i),
\end{aligned} \tag{25}$$

where the equality in $(a)$ is what we are looking for and is similar to the last term in Eq. 20. To this end, let us define $\mathbf{w}_{\backslash \mathcal{V}_{\text{affect}}} = \arg\min_{\mathbf{w}} L_{\backslash \mathcal{V}_{\text{affect}}}(\mathbf{w})$, then we have

$$\begin{aligned}
0 &= \nabla L(\mathbf{w}_{\backslash \mathcal{V}_{\text{affect}}}) - \sum_{i \in \mathcal{V}_{\text{affect}}} \nabla \ell(\mathbf{w}_{\backslash \mathcal{V}_{\text{affect}}}^\top [\mathbf{P}\mathbf{X}]_i, y_i) \\
&\approx \left[ \nabla L(\mathbf{w}_\star) - \sum_{i \in \mathcal{V}_{\text{affect}}} \nabla \ell(\mathbf{w}_\star^\top [\mathbf{P}\mathbf{X}]_i, y_i) \right] \\
&\quad + \left[ \nabla^2 L(\mathbf{w}_\star) - \sum_{i \in \mathcal{V}_{\text{affect}}} \nabla^2 \ell(\mathbf{w}_\star^\top [\mathbf{P}\mathbf{X}]_i, y_i) \right] (\mathbf{w}_{\backslash \mathcal{V}_{\text{affect}}} - \mathbf{w}_\star) \\
&\underset{(a)}{=} \left[ -\sum_{i \in \mathcal{V}_{\text{affect}}} \nabla \ell(\mathbf{w}_\star^\top [\mathbf{P}\mathbf{X}]_i, y_i) \right] + \left[ \nabla^2 L(\mathbf{w}_\star) - \sum_{i \in \mathcal{V}_{\text{affect}}} \nabla^2 \ell(\mathbf{w}_\star^\top [\mathbf{P}\mathbf{X}]_i, y_i) \right] (\mathbf{w}_{\backslash \mathcal{V}_{\text{affect}}} - \mathbf{w}_\star).
\end{aligned} \tag{26}$$

As a result, we can approximate $\mathbf{w}_{\backslash \mathcal{V}_{\text{affect}}}$ by

$$\mathbf{w}_{\backslash \mathcal{V}_{\text{affect}}} = \mathbf{w}_\star + \left[ \nabla^2 L(\mathbf{w}_\star) - \sum_{i \in \mathcal{V}_{\text{affect}}} \nabla^2 \ell(\mathbf{w}_\star^\top [\mathbf{P}\mathbf{X}]_i, y_i) \right]^{-1} \left[ \sum_{i \in \mathcal{V}_{\text{affect}}} \nabla \ell(\mathbf{w}_\star^\top [\mathbf{P}\mathbf{X}]_i, y_i) \right], \tag{27}$$

where both Hessian and gradient are computed on the original graph before node deletion.

## F  LINEAR-GNN IS ALMOST AS POWERFUL AS NON-LINEAR COUNTERPARTS

> *Please notice that this supplementary section is optional. Not reading this section will not affect your understanding of other parts of this paper.*

In this section, we summarize recent studies Wei et al. (2022); Wang & Zhang (2022) shows that linear-GNNs are almost as expressive as its non-linear counterparts. Although two papers study from different perspective (i.e., Wei et al. (2022) studies from Bayesian inference and Wang & Zhang (2022) studies from spectral neural network), they all lead to a similar conclusion that linear-GNNs are almost as expressive as its non-linear counterparts under some assumption on the node features and graph structure properties.

### F.1 FROM BAYESIAN INFERENCE PERSPECTIVE

Wei et al. (2022) compares linear-GNN and non-linear GNN from the Bayesian inference perspective. They consider binary node classification where the graph is randomly generated by contextual stochastic block models (CSBM). They measure the success of node classification by signal-to-noise ratio (SNR). They show that under some assumptions on the CSBM, the SNR of non-linear GNN is in the same order as linear-GNN.

More specifically, Wei et al. (2022) assumes the random graphs are generated by contextual stochastic block models (CSBM), where each node has a feature vector $\mathbf{x}_i \in \mathbb{R}^m$ and a binary category label $y_i \in \{+1, -1\}$. Let us suppose $\mathcal{G}(\mathcal{V}, \mathcal{E})$ is an graph generated by $\mathrm{CSBM}(n, p, q, \mathbb{P}_{+1}, \mathbb{P}_{-1})$ with the following processes:

- (Generate node labels) For each node $v_i \in \mathcal{V}$, we randomly sample the label from $y_i \in \{+1, -1\}$, where $n = |\mathcal{V}|$ is the number of nodes.
- (Generate graph structure) If any two nodes have the same label $y_i = y_j$, then with probability $p$ we add edge $(v_i, v_j)$ to the edge set $\mathcal{E}$. Otherwise, with probability $q$ we add edge $(v_i, v_j)$ to the edge set $\mathcal{E}$.
- (Generate node features) If a node $y_i = +1$, then its feature vector $\mathbf{x}_i$ is sampled from $\mathbb{P}_{+1} = \mathcal{N}(\boldsymbol{\mu}_{+1}, \mathbf{I}/m)$. Otherwise, if a node $y_i = -1$, then its feature vector $\mathbf{x}_i$ is sampled from $\mathbb{P}_{-1} = \mathcal{N}(\boldsymbol{\mu}_{-1}, \mathbf{I}/m)$.

After that, Wei et al. (2022) formulates non-linear GNN and linear-GNN under the context of Bayesian inference, where the optimal propagation is derived from max-a-posterior estimation. To classify a node $v_i$, the optimal non-linear propagation is defined as

$$\mathcal{P}_i = \psi(\mathbf{x}_i) + \sum_{j \in \mathcal{N}(v_i)} \phi(\psi(\mathbf{x}_j); \log(p/q)), \tag{28}$$

where $\psi(\mathbf{x}) = \log(\mathbb{P}_{+1}(\mathbf{x})/\mathbb{P}_{-1}(\mathbf{x}))$ and $\phi(\psi, \log(p/q)) = \mathrm{ReLU}(\psi + \log(p/q)) - \mathrm{ReLU}(\psi - \log(p/q)) - \log(p/q)$. Similarly, for linear-GNN, the optimal linear propagation is defined as

$$\mathcal{P}_i^l(\alpha) = \psi'(\mathbf{x}_i) + \alpha \sum_{j \in \mathcal{N}(v_i)} \psi'(\mathbf{x}_j), \tag{29}$$

where $\psi'(\mathbf{x}) = m \times \left( \langle \boldsymbol{\mu}_{+1} - \boldsymbol{\mu}_{-1}, \mathbf{x} \rangle - (\|\boldsymbol{\mu}_{+1}\|_2^2 - \|\boldsymbol{\mu}_{+1}\|_2^2)/2 \right)$ and $\alpha$ is a parameter to balance information from the root node and its neighbors.

The minimal Bayesian mis-classification error is measured by signal-to-noise ratio (SNR), which is defined as $\rho$ for non-linear GN and $\rho_l$ for linear-GNN,

$$\rho = \frac{(\mathbb{E}[\mathcal{P}_i | y_i = +1] - \mathbb{E}[\mathcal{P}_i | y_i = -1])^2}{\mathrm{variance}(\mathcal{P}_i | y_i = +1)}, \ \rho_l = \max_\alpha \frac{(\mathbb{E}[\mathcal{P}_i^l(\alpha) | y_i = +1] - \mathbb{E}[\mathcal{P}_i^l(\alpha) | y_i = -1])^2}{\mathrm{variance}(\mathcal{P}_i^l(\alpha) | y_i = +1)}. \tag{30}$$

They make the following assumptions on the random graph generator CSBM. More specifically, Wei et al. (2022) assumes the graph structure generated by $p, q$ is neither too strong (e.g., $p \to 1$ and $q \to 0$) or too weak (e.g., graph is too sparse) in Assumption 1 assumes feature generation distributions for positive nodes and negative nodes are not too different.

**Assumption 1** (Assumption on graph structure). *Let us define $\mathcal{S}(p, q) = (p - q)^2/(p + q)$. They assume no very weak graph structure information $\mathcal{S}(p, q) = \omega_n \left( (\log n)^2/n \right)$ and no very strong graph structure information $\mathcal{S}(p, q) \not\to |p - q|$.*

**Assumption 2** (Assumption on node features). *Recall that $\boldsymbol{\mu}_{+1}$ is the mean of positive node feature distribution and $\boldsymbol{\mu}_{-1}$ is the mean of negative node feature distribution. Then, we assume $\sqrt{m}\|\boldsymbol{\mu}_{+1} - \boldsymbol{\mu}_{-1}\|_2 = \mathcal{O}_n(1)$*

Then, Wei et al. (2022) has the following conclusion on the SNR of linear-GNN and non-liear GNN. In particular, they show that non-linear GNN behaves similar to the linear-GNN as their SNRs are in the same order. In other word, under some assumption on graph structure and node features, the linear-GNN could be as expressive as non-linear GNN.

**Theorem 1** (Theorem 2 part 1 of Wei et al. (2022)). *If CSBM satisfy Assumption 1 and Assumption 2, we have $\rho_r = \Theta_n(\rho_l)$.*

**When non-linearity is helpful?** Besides, they show that non-linearity is helpful only if the Assumption 2 does not hold. In other word, if the mean of the positive and negative node feature sampling distribution is different enough $\sqrt{m}\|\boldsymbol{\mu}_{+1} - \boldsymbol{\mu}_{-1}\|_2 = \omega_n(1)$, then $\rho_r = \omega_n(\rho_l)$.

### F.2    FROM SPECTRAL NEURAL NETWORK PERSPECTIVE

Wang & Zhang (2022) shows that linear-GNNs could produce arbitrary predictions under mild conditions on the Laplacian and node features, without relying on the non-linearity in MLP. The expressive power of linear-GNNs mainly comes from its weighted combination of multi-hop graph convolution operators.

Let us define $\mathbf{L} \in \mathbb{R}^{n \times n}$ as the Laplacian matrix in spectral GNNs, where $\mathbf{U}$ is the eigenvectors of $\mathbf{L}$ and $\boldsymbol{\Lambda}$ is the diagonal matrix of eigenvalues. They make the following assumptions on $\mathbf{L}$.

**Assumption 3** (Assumption on $\mathbf{L}$). *No eigenvalues of $\mathbf{L}$ are identical.*

Let us denote $\mathbf{X} \in \mathbb{R}^{n \times d}$ as the node features and $\tilde{\mathbf{X}} = \mathbf{U}\mathbf{X}$ as the graph Fourier transform of node features $\mathbf{X}$. They make the following assumption on $\tilde{\mathbf{X}}$.

**Assumption 4** (Assumption on $\tilde{\mathbf{X}}$). *No rows of $\tilde{\mathbf{X}}$ are zero vector.*

Given any target function $\mathbf{z} = f(\mathbf{L}, \mathbf{X}) \in \mathbb{R}^{n \times 1}$ we want to approximate via linear-GNN. Wang & Zhang (2022) shows that there exists a linear-GNN can approximate function $f(\mathbf{L}, \mathbf{X})$ arbitrary close if Assumption 3 and Assumption 4 hold.

**Theorem 2.** *Let us define $g_{\boldsymbol{\alpha},\mathbf{w}}(\mathbf{L}, \mathbf{X}) = \sum_{\ell=1}^{k} \alpha_\ell \mathbf{L}^\ell \mathbf{X}\mathbf{w}$ as the linear-GNN and $f$ is the target function we want to approximate. Under the Assumption 3 and Assumption 4, there is always exists a set of $\boldsymbol{\alpha}^\star \in \mathbb{R}^k, \mathbf{w}^\star \in \mathbb{R}^d$ such that $g_{\boldsymbol{\alpha}^\star,\mathbf{w}^\star}(\mathbf{L}, \mathbf{X}) = f(\mathbf{L}, \mathbf{X})$.*

In practice, Wang & Zhang (2022) found Assumption 3 and Assumption 4 are very likely to hold on the real-world datasets.

**When non-linearity is helpful?** They show that adding non-linear MLP to linear-GNNs could alleviate the conditions on node features (i.e., Assumption 4) because the output of multi-layer neural network are very likely to satisfy this condition. However, adding non-linearity will not necessarily improve its expressive power if the conditions are already satisfied in the first place.

