# OpenReview forum: "GraphEditor: An Efficient Graph Representation Learning and Unlearning Approach"
_ICLR.cc/2023/Conference — Submitted to ICLR 2023_

### Official Review · Reviewer_FJsd · 2022-10-19

**Confidence:** 4
**Correctness:** 2
**Technical Novelty And Significance:** 3
**Empirical Novelty And Significance:** 2
**Recommendation:** 5

**Clarity, Quality, Novelty And Reproducibility:**

- Clarity: The paper does not emphasize the key assumption that GraphEditor only applies to ridge regression. Also, please clearly specify whether ridge regression or cross-entropy loss is used in the experiment. Overall, I feel the paper is misleading.

- Quality: For ridge regression, the results and proofs are correct. However, the main task considered in the paper is the classification problem, where logistic loss or cross-entropy loss should be considered.

- Novelty: The novelty of the paper is limited. The analysis on ridge regression is quite standard.

- Reproducibility: The experiment code is attached. However, the readme file is empty and does not specify the environment and package dependency to run the code.


**Strength And Weaknesses:**

### Strengths

- GraphEditor demonstrates great empirical results in terms of efficiency and test accuracy.

- The related works are extensive.

### Weaknesses

- The proposed approach is only proved to be an exact unlearning method for the ridge regression problem on graphs. Yet the authors claim to study the node classification problem, which usually uses either logistic regression or cross-entropy loss.

- The overall writing of the paper is quite misleading, assuming the first weakness is valid.

- The GraphEditor ***cannot*** extend to non-linear models such as general GNNs. The authors only use pretrained GNNs (on private datasets) as a ***fixed*** feature extractor. The learning and unlearning processes apply only to linear classifiers on public datasets.

- The setting of testing the unlearning ability of approximate unlearning methods is unclear. Do the authors use ridge regression or cross-entropy loss?

### Detail comments

Graph unlearning is a very important problem that is just recently been aware by the community. I was very excited at the beginning when I see the claims and fascinating experimental results. Unfortunately, I found some critical issues regarding the method itself which I discussed below. I am more than happy to change my mind if the authors can correct me.

The major and most critical issue is that the paper overclaims its contribution and ignores the important assumption of GraphEditor when conducting the experiments. Note that all analyses pertaining to GraphEditor rely on the structure of the ***ridge regression problem*** (on graphs), see Section 4.1. For example, the key part of the method is to show that the resulting unlearned weights $W_{upd}$ is ***identical*** to the one retraining from scratch $W^{u}_\star$ (when the training algorithm is guaranteed to converge to the unique optimum). The authors rely on the closed-form solution and the quadradic nature of the problem to establish their deletion and update steps (Section 4.2). However, the real task that we aim to learn is the classification problem, which usually adopts logistic loss or cross-entropy loss. Note that the authors also claim to study the classification problem and conduct experiments on node classification tasks. Unfortunately, the analysis and guarantee established in Section 4.2 cannot generalize to even logistic loss. Hence, the proposed GraphEditor unlearning approach is ***not*** an exact unlearning method in general graph representation learning and I feel the authors overclaim their contributions.

Note that in the experiments, the authors also seem to leverage cross-entropy loss (softmax) as stated in Section 5.2. Thus, the GraphEditor is not guaranteed as an exact unlearning method in theory throughout the experiments in Section 5. Note that the proposed deleted data replay test is not sufficient to validate whether the model is exactly unlearned or not. The only criterion to validate whether GraphEditor is an exact unlearning method is to check if its resulting weights after unlearning $W_{upd}$ is ***identical*** to the one retraining from scratch $W^u_\star$ in distribution. Unfortunately, it is unlikely to verify whether $W_{upd}$ and $W^u_\star$ are identical in distribution in practice. This is why theoretical guarantees are very important and critical in the context of machine unlearning, especially when one claims a method to be an exact unlearning method.

Interestingly, if we consider the special case where all nodes in the graph are isolated (i.e. the propagation matrix is merely an identity matrix), the situation falls back to the standard machine unlearning problem. Consider again the case of the classification task, which is the problem studied by [1] under the approximate unlearning criteria. Apparently, GraphEditor is also claimed to be an exact unlearning method in this scenario. Due to the superior time complexity demonstrated by the authors, why does one even need approximate unlearning approaches such as the one proposed in [1]? Notably, Guo et al. [1] already mentioned that when the problem is ridge regression (i.e. least-squared loss with $\ell_2$ regularization), then the approach therein is an exact unlearning method (Section 3.2). This also emphasizes the fact that ridge regression is less interesting for machine unlearning.

The other major issue is the claim of GraphEditor being able to generalize to non-linear graph models such as GNNs. Note that the authors only adopt the approach proposed in [1] and [2], which simply pretrain non-linear models on private datasets as ***fixed*** feature extractors. The learning and unlearning processes apply only to linear classifiers on public datasets. I think the statements in the abstract and the introduction on GraphEditor can be generalized to non-linear models such as GNNs are overclaimed and misleading.

I would suggest the authors emphasize that GraphEditor only works with ridge regression problems, albeit this greatly diminishes the contribution of the paper. Also, I would suggest the authors to tone down the claim on the generalization ability of GraphEditor to non-linear models.

### References

[1] Certified data removal from machine learning models, Guo et al., ICML 2020.

[2] Mixed-privacy forgetting in deep networks, Golatkar et al., CVPR 2021.


**Summary Of The Paper:**

The paper studies the problem of machine unlearning on graphs. The authors claim an exact unlearning method (GraphEditor) that supports node/edge deletion/addition and node feature update. The experiments demonstrate the efficiency, high performance, and ability of unlearning of the method.

**Summary Of The Review:**

The authors proposed GraphEditor and claim it to be an exact unlearning method for general graph representation learning, especially node classification problems. However, the unlearning guarantee relies on the problem being the ridge regression problem and does not generalize to the other popular classification loss such as logistic loss and cross-entropy loss. The experiments seem to leverage cross-entropy loss which makes the result meaningless. I feel the contribution is overclaimed and the assumption on ridge regression needs to be emphasized. Otherwise, the paper is misleading in its current form. I might misunderstand the paper and I am more than happy to change my mind if the authors can correct me.

================post rebuttal===========================

I thank the authors for their effort in addressing my major concern. Now I agree that GraphEditor is indeed an exact unlearning method. Thus I increase my rating accordingly. However, some new concerns arise (see my follow-up comments below). Overall, I still think the manuscript should be further improved before publication, especially in its clarity.

---

> ### Author Response · Authors · 2022-11-13
> **Response to Reveiwer FJsd (Part 1)**
>
> **Q1. GraphEditor is not an exact unlearning method as claimed by the authors.**
>
> A1. In the following, we will explain why GraphEditor is exact unlearning and why its solution is identical to re-training.
>
> We first consider the first three steps in Algorithm 1, i.e., $\mathbf{W}=\text{find\\_W}(\mathbf{A}, \mathbf{X})$, $\mathbf{W}_\text{unlearned} = \text{remove\\_data}(\text{add\\_data}(\mathbf{W}))$. Since GraphEditor applies the Sherman–Morrison–Woodbury formula (Lemma 3) onto the closed-form solution of the ridge regression, it guarantees the output after remove\_data and add\_data is identical to the solution obtained by re-training with find\_W on the remaining data, i.e., $\mathbf{W}_\text{unlearned} = \mathbf{W}_\text{remain}= \text{find\\_W}(\mathbf{A}_\text{remain}, \mathbf{X}_\text{remain})$.
>
> Then, we consider fine-tuning with cross-entropy loss (the last step in Algorithm 1). Since we fine-tuning on $\mathbf{W}_\text{unlearned}$, the weight parameters guarantee not carrying any information on the deleted node. Besides, since the starting point of our fine-tuning process $\mathbf{W}_\text{unlearned} = \mathbf{W}_\text{remain}$, then we know fine-tune on $\mathbf{W}_\text{unlearned}$ and $\mathbf{W}_\text{remain}$ for the same number of iterations should give us the solution. Therefore GraphEditor is exact unlearning and its solution is identical to re-training.
>
> Please notice that we have empirically verified whether these values are identical in Figure 2.
>
> **Q2. Guo et al. [1] with ridge regression is identical to GraphEditor. Ridge regression is less interesting for machine unlearning.**
>
> A2. In the following, we explain why GraphEditor is different from [1] with Ridge regression by first ignoring the node dependency and then considering the node dependency.
>
> **When ignoring the node dependency.** Please notice using [1] with Ridge regression is different from GraphEditor and has different computation complexity. In particular, [1] has complexity grows linear w.r.t. remaining data size, however, GraphEditor is independent with remaining data size. For example, when applying Newton's method in [1] with Ridge regression we have
>
> $$
>  \mathbf{W}_\star^u = \mathbf{W}_\star - \big[ \nabla^2 \mathcal{L}_\text{Ridge}(\mathbf{W}; \tilde{\mathbf{X}}) \big]^{-1} \nabla \mathcal{L}_\text{Ridge}(\mathbf{W}; \tilde{\mathbf{X}}, \mathbf{Y})$$
>
> $$= \mathbf{W}_\star - \big[\tilde{\mathbf{X}}^\top \tilde{\mathbf{X}} + \lambda \mathbf{I} \big]^{-1} \big( (\tilde{\mathbf{X}}^\top \tilde{\mathbf{X}} + \lambda \mathbf{I}) \mathbf{W}_\star - \tilde{\mathbf{X}}^\top \mathbf{Y} \big)
> $$
>
> $$ = (\tilde{\mathbf{X}}^\top \tilde{\mathbf{X}} + \lambda \mathbf{I})^{-1} \tilde{\mathbf{X}}^\top \mathbf{Y} $$
>
> $$=  {\arg\min}_{\mathbf{W}}~ \mathcal{L}_\text{Ridge}(\mathbf{W};\tilde{\mathbf{X}},\mathbf{Y})$$
>
>
> The computation cost is proportional to the number of remaining data because they need to re-compute the gradient $\nabla \mathcal{L}_\text{Ridge}$ and Hessian $\nabla^2 \mathcal{L}_\text{Ridge}$ on the remaining data.
> On the other hand, our computation complexity is only related to the number of nodes to delete (irrelevant to the number of remaining data $R$).
> For more details please refer to "challenges in graph unlearning" in Section 3 and Appendix C4.
>
> **If considering the node dependency.**
> The node dependency issue makes employing [1] even non-trivial because it needs to also remove the effect of all neighbor nodes, please refer to Section 2's paragraph "Approximate machine unlearning" and Appendix E for details. Please notice that our analysis in Appendix E is general and holds for both logistic and ridge regression.
>
> **Square loss is not interesting.**
> We believe square loss could still be attractive for research. For example, the NeurIPS21 paper [2] shows that the solution obtained by Ridge regression can be used as initialization of a non-linear deep model for efficient training; the NeurIPS22 paper [3] shows that the solution obtained by square loss can be used to overcome the performance degradation issue on heterophily graphs. Similarly, we study square loss for efficient and exact graph representation unlearning.
>
> [1] Certified Data Removal from Machine Learning Models https://arxiv.org/abs/1911.03030
>
> [2] Implicit SVD for Graph Representation Learning https://arxiv.org/abs/2111.06312
>
> [3] Simplified Graph Convolution with Heterophily https://arxiv.org/abs/2202.04139

---

> > ### Author Response · Authors · 2022-11-13
> > **Response to Reviewer FJsd (Part 2)**
> >
> > Q3. The statements in the abstract and the introduction on GraphEditor can be generalized to non-linear models such as GNNs are overclaimed and misleading. I would suggest authors tone down the claim on the generalization ability of GraphEditor to non-linear models.
> >
> > A3: Sorry for the confusion. The main focus of this paper is still on linear-GNN as we have highlighted our paper as "linear-GNN" in the paper's keywords. To emphasize the study of linear-GNN is still interesting and important, we take a full section in Appendix F to show that linear-GNN has almost the same expressive power as non-linear.
> > We introduce the non-linearity extension to show GraphEditor could still be used with non-linear models (similar to [1] and [2])
> >
> > Meanwhile, as suggested by the reviewer, we double-checked our presentation in the abstract and introduction. In fact,
> >
> > - we did not mention generalization to non-linear in abstract
> > - we only mention it once in the last paragraph of the introduction "*We introduce a graph representation learning and unlearning approach GraphEditor on linear-GNNs ... To improve the scalability and expressiveness, we introduce subgraph sampling and the non-linearity extension of GraphEditor*".
> >
> >
> > To avoid potential confusion, we highlight linear-GNN in the abstract.
> >
> > [1] Mixed-Privacy Forgetting in Deep Networks https://arxiv.org/abs/2012.13431
> >
> > [2] Eternal Sunshine of the Spotless Net: Selective Forgetting in Deep Networks https://arxiv.org/abs/1911.04933

---

> > > ### Comment · Reviewer_FJsd · 2022-11-22
> > > **Follow-up questions**
> > >
> > > I thank the detailed response by the authors, my concerns pertaining to ridge regression is addressed and resolved. I increase my rating accordingly. However, it raises some new concerns for me now, which I explained as follows.
> > >
> > > According to my current understanding, the authors treat the ridge regression solution (close-form) as a ***initialization*** and then train the model on the cross-entropy loss for datasets after removal. Hence, GraphEditor is indeed an exact unlearning method. However, as I also mentioned in my comments, we usually use cross-entropy loss for the node classification task. Hence, the ***retrain*** model baseline should be fully trained with cross-entropy loss. In section 5.1, the authors do not compare the accuracy with this baseline and I wonder why. The results should better demonstrate what should we pay to enable unlearning (or is it for free)?
> > >
> > > On the other hand, I also have questions about the experiment setting of Table 1, the ***before*** rows. Do GraphEditor (no fine-tuning) results obtain with ridge regression loss and other baseline results obtain with cross-entropy loss? If so, it is quite surprising that the ridge regression solution can be on par or better with the cross-entropy loss solution. Do the authors just simply apply softmax on the output of GraphEditor during prediction? On the other hand, what are the privacy parameters $(\epsilon,\delta)$ setting for the approximate unlearning methods such as Influence and Fisher? Note that larger $\epsilon$ allows us to choose smaller noise injected into the model, which ideally makes the models have better accuracy. Also, do the authors tune the regularization parameter $\lambda$ for approximate unlearning methods similar to GraphEditor? I could not find a detailed explanation about this, it would be great if the authors can mention where they are in the appendix (if they were there).
> > >
> > > Finally, it seems to me that the main idea of GraphEditor (i.e., using ridge regression solution and fine-tuning on cross-entropy loss) can also be applied to the case without graphs. I wonder how it performs in the case without graphs (i.e., similar to the setting in Guo et. al.). Does this method simply outperform Guo et. al. in terms of accuracy and complexity while ensuring exact unlearning?

---

> > > > ### Author Response · Authors · 2022-12-09
> > > > **Feedback to reviewer**
> > > >
> > > > Thank you for raising this question. Experiment in Section 5.1 is mainly designed to verify whether unlearning method can truly unlearning and whether their method would hurt model performance, therefore we only focus on the unlearning methods proposed in existing paper. In terms of the performance of re-training with cross entropy loss vs GraphEditor, we would like to note that re-training with cross entropy loss theoretically has the same solution as (GRAPHEDITOR + Fine-tune) because linear-GNN is convex and has the unique optimal solution.
> > > >
> > > > For the second question, all baselines are using logistic regression (following their official implementation). The reason why GraphEditor has higher accuracy than Influence and Fisher because these methods need to use different regularization term $\lambda$ to balance the performance before and after unlearning. More specifically, these two methods need Hessian inverse to estimate the approximate unlearned solution, and the norm of the Hessian inverse is upper bounded by
> > > > $||[ \nabla^2 \ell(\mathbf{w})]^{-1}||_2 \leq 1/\lambda$.
> > > > When $\lambda$ is small, we are facing the gradient exploding issue where the gradient norm is an order of magnitude larger than the weight norm, such that the unlearned model cannot generate meaningful predictions. On the other hand, a larger $\lambda$ will hurt the model’s learning ability and results in a poor performance before unlearning. We select their $\lambda$ from $\\{0, 10^{-4},\ldots, 10^{-6}\\}$ for experiments. The differential privacy hyper-parameters we follow their official implementation. Besides, we would like to notice that the regularization term is not limited in GraphEditor and we can select a smaller regularization (e.g., $10^{-6}$ or 0) for better performance. On the other hand, GraphEditor has higher accuracy than GraphEraser because graph partitioning causes data heterogeneity and a lack of training data for each subgraph model. We will incorporate the above discussion into the updated draft and highlight them in the main text.
> > > >
> > > > Our paper mainly focus on graph structured data, but we would like to test it on non-structured data as well.

---

### Official Review · Reviewer_5Djq · 2022-10-24

**Confidence:** 3
**Correctness:** 2
**Technical Novelty And Significance:** 2
**Empirical Novelty And Significance:** 2
**Recommendation:** 3

**Clarity, Quality, Novelty And Reproducibility:**

The writing of this paper can be further improved. Here are some aditional minor points:
- Briefly introduce unlearning in the abstract, to make it more clear.
- A pipeline figure is better than algorithm (pseudocode) for illustration.


**Strength And Weaknesses:**

Strengths:
- Graph unlearning is an interesting problem, especially in the large-scale graph setting.
- I’m not an expert on graph unlearning, but the experiments seem to prove the effectiveness of GraphEditor algorithm. I’m open to hear comments from other reviewers on this.


Weakness:
- The new metric, “deleted data reply test”, is interesting, but the logic is not clear. A better logic is that, what this measure is about, and how existing methods fail on this method. Then it can be more reasonable to introduce the GraphEditor algorithm. The current logic is that, in Sec 3, the authors propose a metric for data removal guarantee, then without further explanation, goes directly to introducing the algorithm Sec 4. This logic is not clear to the readers.

- In Sec 4, The three main functions can be renamed, e.g., find-W is actually the GNN learning. Similarly for remove_data() and add_data(), where the namings are confusing.

- For Lemma 1, there are some places I want to confirm with the authors.
  - Where is L defined in Lemma 1? I can only find it implied in Sec 3, if it’s the number of GNN layers. Can authors confirm this?
  - If so, then Lemma 1 is straightforward for linear GNN. It is essentially saying that for L-layer linear GNN, the effect of unlearning is restricted in the 2L-hop neighborhood.
  - Now the authors are considering a simplieid case without activation function. What if the non-linear activation functions are considered? I didn’t find this explicitly discussed in Sec 4.4. Sec 4.2 makes sense because the parameters can be learned in a closed-form, which is not the case for Sec 4.4, and authors say just replacing the linear GNN with non-linear GNN is not convincing.


- Another question on the backbone modeling. So this work is focusing on the GNN. But actually the most recent works [1,2] find that MLP can achieve similar performance as GNN. Then the unlearning can much simpler in that case. Does author have any comment on this?


**Summary Of The Paper:**

This paper studies graph unlearning, a problem on how to quickly adapt the GNN after removing/adding a subgraph like nodes and edges. This work proposes GraphEditor, an efficient learning algorithm that does not need GNN retraining, with motivation on the linear GNN.


**Summary Of The Review:**

My biggest concern is on the generalization of Lemma 1 to non-linear GNN. If it's still an open challenge, as mentioned by the authors, then maybe Lemma 1 can be skipped, and authors can consider telling the story from another direction.

---

> ### Author Response · Authors · 2022-11-13
> **Response to Reviewer  5Djq**
>
> **Q1. Consider first introducing the "deleted data reply test" measure, then explain how existing methods fail on this method, and finally introduce GraphEditor. The authors propose a metric for data removal guarantee, then without further explanation, goes directly to introducing the algorithm Sec 4. This logic is not clear to the readers.**
>
> A1. Thank you for the suggestions and sorry for the inconvenience. Please notice that the "deleted data reply test" is just one of the evaluations we used in this paper. We also consider many other evaluation methods in Section 5 and Appendix A. We believe emphasizing too much on the "deleted data reply test" before introducing our algorithm makes our paper sounds like an algorithm that is only designed to solve the "deleted data reply test". Moreover, we would like to highlight that the "deleted data reply test" is mainly designed to show approximate unlearning might not unlearn and jeopardizes privacy, while GraphEditor is exact unlearning which guarantees perfect unlearning.
>
> In fact, we did not introduce the "deleted data reply test" at the end of Section 3. Instead, we said in the last point of Section 3 that *"it is important to show from the algorithm itself that the sensitive information can be perfectly removed, ..., we propose GraphEditor that enjoys a low computation cost with data removal guarantees."* which leads us to our method section in Section 4. We believe this is the correct logic to first mention the limitation of approximate unlearning and then introduced GraphEditor as exact unlearning.
>
> **Q2. Is L the number of GNN layers? If so, then Lemma 1 is straightforward for linear GNN. What if the non-linear activation functions are considered? authors say that just replacing the linear GNN with a non-linear GNN is not convincing.**
>
> A2: Yes, L is the number of GNN layers. Please notice that since GraphEditor is applied to the feature representation computed by the feature extractor (i.e., linear-GNN or non-linear GNNs), it is not affected by whether using non-linearity or not. For linear-GNN the feature representation $\mathbf{X} = \mathbf{P}^L \mathbf{H}^{(0)}$ while for non-linear GNN is $\mathbf{X} = \sigma(\mathbf{P} \sigma(\mathbf{P} \mathbf{H}^{(0)} \mathbf{W}^{(1)})\mathbf{W}^{(2)} )$. Please kindly refer to Figure 1 and Appendix D (proof of Lemma 1) for details. For example, the proof of Lemma 1 also holds after considering the activation function because using the activation function will not affect which nodes are affected (the reviewer can refer to Figure 1 for an illustration).
>
>
> **Q3. recent works [1,2] find that MLP can achieve similar performance as GNN. Then the unlearning can be much simpler in that case. Does the author have any comment on this?**
>
> A3. Using MLP instead of GNN will reduce the graph representation unlearning problem to the general machine unlearning problem which has been studied in existing works as we summarized in Section 2. Compared to graph representation unlearning, they no longer face the node dependency issue.
> In this paper, we focus on graph representation unlearning to take the node dependency into consideration, because we believe neighbor information could still be important in most cases on a graph.

---

### Official Review · Reviewer_BkPA · 2022-10-26

**Confidence:** 3
**Correctness:** 3
**Technical Novelty And Significance:** 3
**Empirical Novelty And Significance:** 3
**Recommendation:** 6

**Clarity, Quality, Novelty And Reproducibility:**

The presentation is in general clear. Please address some of the points raised in weaknesses for better clarity.

The paper ensembles a lot of materials to solve a very important but relatively under-explored problem. Though some strong assumption (e.g. linearity) limits its technical challenge and novelty, its empirical evaluation is convincing, and it is a good complement to the existing literature.

Though I do not check the proof and codes in great detail, I believe it is technically sound.

**Strength And Weaknesses:**

Strengths:

- GraphEditor enables the influence of deleted node to have closed form and can be explicitly computed, which theoretically guarantees the exact unlearning.

- The proposed approach flexibly supports various scenarios including node/edge updating and also introduces subgraph sampling to address scalability issues.

Weaknesses:

- The linearity assumption limits the technical contribution. the authors do provide some discussions on why linear GNN may be expressive enough and existing literature also uses linearity. To better understand the technical contribution of this paper, please clarify more on the essential barrier when transferring proposed approach and its theoretical guarantee to nonlinearity, i.e. why it is nontrivial in its current form and how potentially we could make progress. without such discussion, there is a significant logic gap from Sec 4.2 to Sec 4.4.

- in Section 4.1, the paper proposes to use ridge regression to initialize and fine-tune using a different task-related loss term (e.g. cross-entropy loss) with a small number of iterations.

     - Is there potential issue that the unlearning is specific to a certain loss structure (ridge loss) while is insufficient for another.
     - Please discuss whether the given guarantee is generalizable to cross entropy loss, where the experiments are mainly about.

- does the competing baseline also uses some trick like fine-tuning to boost performance in experiments, to ensure a fair comparison?

- is there a guarantee that fine-tuning from such initialization is faster than retrain to get comparable performance?


**Summary Of The Paper:**

This paper proposes a computationally efficient exact unlearning approach called GraphEditor on graph neural networks. The main challenge of unlearning on graph data is the interconnection between neighboring nodes makes elimination of influence from a given deleted node nontrivial. This paper considers a linearized GNN, in which the influence of a given node on the model weight could be explicitly written out and computed. GraphEditor is thus guaranteed to remove all information related to deleted nodes. The author also introduce a "deleted data replay test" to validate whether information is actually forgotten.

**Summary Of The Review:**

This paper is technically solid and provides a good contribution to graph unlearning problem. Though the theoretical contribution is restricted by the linearity constraint, it serves as a good starting point. I enjoy reading the paper and vote for acceptance.

---

> ### Author Response · Authors · 2022-11-13
> **Response to Reviewer BkPA**
>
> **Q1. Why it is non-trivial to employ unlearning in the non-linear model? How potentially we could make progress?**
>
> A1. Due to the high non-convexity and complicated optimization landscape of deep non-linear models, it is still not clear how we can unlearn and how to verify unlearn.
>
> - (how to unlearn is not clear) Our current understanding of non-linear neural networks is still limited, it is almost impossible to trace the influence of the deleted nodes on the weight parameters in a deep neural network model due to its non-convexity. However, if the model is convex (e.g., the linear-GNN model), then we could explicitly trace the effect of the deleted data.
>
> - (how to verify unlearn is not clear) We do not have a rigorous solution to evaluate unlearning for non-convex models. For example, some existing methods propose to measure the similarity of the weight parameters or the model output to the re-training solution or use a membership inference attack as a proxy for evaluation. Unfortunately, recent studies [1, 2, 3] show that designated attacks could still attack approximate unlearning to learn about private data (e.g., membership inference attack) which leads to some negative results on empirically unlearning guarantee.
>
>     [1] Forget Unlearning: Towards True Data-Deletion in Machine Learning https://arxiv.org/abs/2210.08911
>
>     [2] On the necessity of auditable algorithmic definitions for machine unlearning https://arxiv.org/abs/2110.11891
>
>     [3] Evaluating Inexact Unlearning Requires Revisiting Forgetting https://arxiv.org/abs/2106.04378
>
>
> To make progress on the non-linear model, we have to first carefully study a simpler case to carefully understand how weight parameters are affected after data deletion. We believe that conducting research step by step could help us build a more solid understanding of a new field and avoid non-trivial pitfalls. Therefore, we believe GraphEditor is a good starting point to better understand graph unlearning before moving into deep non-linear GNN models because (1) convexity allows us to explicitly mathematically analyzed in a rigorous manner and (2) linear-GNN is known to be as expressive as deep non-linear GNNs (Appendix F).
>
>
>
> **Q2: GraphEditor is designed by using the closed-form solution of ridge regression. Can we generalize it to cross entropy loss and its there any limitations?**
>
> A2. Thank you for asking. GraphEditor requires us to re-formulate the original GNN training problem as a ridge regression problem to apply Sherman–Morrison–Woodbury formula for exact unlearning. We cannot generalize it to cross-entropy loss.
>
> **Q3: Does baseline use fine-tuning? Is there any guarantee that fine-tuning faster than re-training from scratch?**
>
> A3. Thank you for asking. Our baseline implementation follows the officially released implementation and does not involve fine-tuning. We would like to emphasize that the biggest advantage GraphEditor over the approximate unlearning baseline is that GraphEditor is exact unlearning that could perfectly unlearn the deleted data. Using fine-tuning or not could not help approximate unlearning alleviate this "might not unlearn" issue.
>
> In terms of whether fine-tuning is faster than re-training, we compared the wall-clock time between "GraphEditor+fine-tuning", and "Re-training from scratch" in Table 1 (2nd and 3rd-row blocks) and Table 3 (2nd and 3rd-row blocks). For example in Table 1, "GraphEditor+fine-tuning" takes around 15 sec on OGB-Arxiv but re-training with "GraphEraser (parallel re-training with linear-GNN )" takes more than 2,237 seconds when the number of unlearning split is S=100. Similarly, "GraphEditor+fine-tuning" takes around 118 sec on OGB-Products but re-training with "GraphEraser (parallel re-training with linear-GNN )" takes more than 46,491 seconds when the number of unlearning split is S=100.

---

### Official Review · Reviewer_N3CX · 2022-10-27

**Confidence:** 4
**Correctness:** 3
**Technical Novelty And Significance:** 2
**Empirical Novelty And Significance:** 3
**Recommendation:** 5

**Clarity, Quality, Novelty And Reproducibility:**

Clarity

The paper is clearly written and easy to understand.

Quality

I am worried on the closed form solution derived with Sherman–Morrison–Woodbury formula for multi-row update. I expect the authors to clarify on that. Experiments are with high quality.

Novelty

The idea of inserting extra binary feature to check the unlearning quality is novel and useful

Reproducibility

For replicating the results reported in this submission, it is not for now

**Strength And Weaknesses:**

Strength
1. The proposed approach is simple and efficient, which does not scale with the dataset size
2. The idea of inserting extra binary feature to check the unlearning quality is novel and useful
3. Experiment results are promising

Weaknesses
1. The closed form solution in derived with Sherman–Morrison–Woodbury formula, and the authors only show the derivation for only one row deleted. Maybe I miss something but I do not think it can be directly extend to matrix case (multiple rows) with the same formula.
2. The approach is only for linear GNN, which is very limited. Although the authors argue that other graph unlearning algorithms also requires that for theoretical guarantee, I think in general the other approaches should work for non-linear case without theoretical guarantee. But the proposed work does not work for non-linear case
3. The proposed connection to non-linear GNN makes sense but not practical. In industry use case no users' data should be forbidden to delete, and using public dataset to train the feature extractor could deteriorate the model performance. For section 5.3 the authors still use the same training dataset to train the feature extractor, the statement could be validated better if the authors could find some other dataset to train the feature extractor.

**Summary Of The Paper:**

The paper proposes a graph unlearning approach for linear graph neural networks, with ridge regression as the objective. The authors derive closed form solution for deleting nodes in the graph. For experiments, the authors propose to insert extra binary feature indicating whether the node is to be removed in node features to validate if the unlearned model has prediction power of the extra binary feature. The authors also compare the model closeness of unlearned model re-trained model. Other extensive experiment results are also shown to show the effectiveness and efficiency of the proposed approach. The approach can be extended to non-linear GNN case by pre-training the feature extraction part with non-deletable data and only train a linear classifier.

**Summary Of The Review:**

The paper shows novelty on evaluating the quality of unlearning. However, there are still many limitations of the approach as it only works for linear GNN, I do not recommend to accept it now.

---

> ### Author Response · Authors · 2022-11-13
> **Response to Reviewer N3CX (Part 1)**
>
> **Q1. Can Sherman–Morrison–Woodbury formula be extended from the vector case (Lemma 3) to the matrix case?**
>
> A1: Thank you for raising this question. This formula could be extended to the matrix case (a.k.a. Woodbury matrix identity). Please refer to Section 3.2 of [1]. We updated the Lemma 3 to make this crystal clear in the revised version.
>
> [1] The Matrix Cookbook https://www.math.uwaterloo.ca/~hwolkowi/matrixcookbook.pdf
>
> **Q2. Can we apply existing unlearning methods for the non-linear case if ignoring the theoretical guarantee?**
>
> A2: This is indeed a great question. Please notice that most of the algorithm design of existing unlearning methods are mathematical oriented and requires linear assumption. When these assumptions are violated, we not only cannot guarantee whether the algorithm work but also cannot rigorously verify the effectiveness of unlearning. For example
>
> - Paper [1] unlearns by using first-order Taylor approximation, which requires the solution before unlearning to achieve optimal with zero gradients. This requirement/assumption is impractical on deep non-linear models due to its non-convexity. Unless severely overfitting the training data, achieving zero gradients is impractical on deep non-linear models. Moreover, since [1] requires the solution before unlearning pre-trained by optimizing random noises perturbed loss function, which makes it even impractical to achieve this optimal point.
>
>     [1] Certified Data Removal from Machine Learning Models https://arxiv.org/abs/1911.03030
>
>
> - Papers [2] and [3] only unlearn the last layer of a deep neural network (similar to our Section 4.4). They unlearn by first using Newton's method for approximation and then adding random noise to weight parameters to make model output indistinguishable. It cannot be applied to a deep non-linear model because using one step of Newton's method (i.e., second-order gradient descent) on the non-linear model cannot guarantee approximation due to non-convexity. More importantly, their random noise strategy is derived from linear-model and it cannot extend to non-linear models because of the multi-layer composite effect in the non-linear model.
>
>     [2] Mixed-Privacy Forgetting in Deep Networks https://arxiv.org/abs/2012.13431
>
>
>     [3] Eternal Sunshine of the Spotless Net: Selective Forgetting in Deep Networks https://arxiv.org/abs/1911.04933
>
> - The algorithm in [4] and [5] requires the objective convex to guarantee L-BFGS could convergence, and they conduct experiments using strong-convex objective in practice. Similarly, the algorithm [6] and [7] require the objective as convex or strongly convex. These methods cannot be applied to non-linear models because non-linear models are non-convex.
>
>     [4] DeltaGrad: Rapid retraining of machine learning models https://arxiv.org/abs/2006.14755
>
>     [5] Remember What You Want to Forget: Algorithms for Machine Unlearning https://arxiv.org/abs/2103.03279
>
>     [6] Descent-to-Delete: Gradient-Based Methods for Machine Unlearning https://arxiv.org/abs/2007.02923
>
>     [7] Machine Unlearning via Algorithmic Stability https://arxiv.org/abs/2102.13179
>
> Most importantly, we cannot design a rigorous experiment for evaluation.
> The heuristic-designed unlearning evaluation methods (e.g., comparing the similarly on parameter space or output space, using membership inference attack as a proxy evaluation tool) may have many pitfalls that are hard to discover. For example, recent studies [8, 9, 10] show that approximate unlearning could still be attacked by designated attacks to learn about private data (e.g., membership inference attack) which leads to some negative results on empirically unlearning guarantee.
>
> [8] Forget Unlearning: Towards True Data-Deletion in Machine Learning https://arxiv.org/abs/2210.08911
>
> [9] On the necessity of auditable algorithmic definitions for machine unlearning https://arxiv.org/abs/2110.11891
>
> [10] Evaluating Inexact Unlearning Requires Revisiting Forgetting https://arxiv.org/abs/2106.04378
>
> As an alternative, we choose first to study the unlearning problem in linear-GNNs before jumping into non-linear cases. We believe GraphEditor is a good starting point to better understand graph unlearning before moving into non-linear GNN models because (1) the convexity in linear-GNN allows us to explicitly mathematically analyzed in a rigorous manner and (2) linear-GNN is known to be as expressive as deep non-linear GNNs (Appendix F) and its experimental performance is close or even outperform non-linear GNN (e.g., Table 1, 3, 4, 5, 8). We believe that conducting research step by step could help us build a more solid understanding of a new field and avoid non-trivial pitfalls.

---

> > ### Author Response · Authors · 2022-11-13
> > **Response to Reviewer N3CX (Part 2)**
> >
> > **Q3. The proposed connection to non-linear GNN makes sense but is not practical.**
> >
> > A3. Our Section 4.4 is mainly to demonstrate there exists an opportunity to apply GraphEditor with non-linear GNNs, but our main focus is still on linear-GNNs. That is why we simply call it a "non-linear extension" and only spend very limited space discussing it (1 paragraph to introduce it and  1 paragraph for the experiment).
> > Our experiments in Section 5.3 simply simulate this public-private dataset setting by splitting the original dataset into two parts to demonstrate the possibility of using GraphEditor with non-linear GNN models.

---

### Author Response · Authors · 2022-11-18
**To all reviewers**

Dear Reviewers,

The discussion period is closing tomorrow. We would like to thank you again for your time reading and evaluating our work. We have also answered thoroughly your questions individually and point-by-point. We would appreciate knowing if you have any additional questions, concerns, or clarifications you would like to ask us. If we have addressed your concerns, please, consider revising your score.

Best regards,

The authors

---

### Decision · Program_Chairs · 2023-01-20

**Decision:**

Reject

**Justification For Why Not Higher Score:**

The presentation is a bit misleading and limitation of the current approach are not clearly presented. Paper may be interesting but it needs major rewriting.

**Justification For Why Not Lower Score:**

N/A

**Metareview: Summary, Strengths And Weaknesses:**

The paper presents GraphEditor, a new efficient method for graph representation learning and unlearning. Graph unlearning is an important task with many privacy applications, in this task one is interested in removing the effect of a node from the pre-trained graph representation learning of a graph efficiently.

The presented method is interesting and it presents nice theoretical and experimental results for linear GNN.

The main drawback of the paper is that almost all the GNN used are non linear and so the current result has limited applicability. In addition, it is also hard to imagine that the current approach could be generalized to more complex network structure because it heavily relies on the linear structure of the problem.

In addition, the theoretical result is not particularly novel or surprising in the linear setting.

Overall, the paper contains some interesting ideas but the paper does not meet the ICLR acceptance bar.